# Delving into RL for Image Generation with CoT: A Study on DPO *vs.* GRPO

**Chengzhuo Tong**[*4], **Ziyu Guo**[*†1], **Renrui Zhang**[*‡2], **Wenyu Shan**[*3], **Xinyu Wei**[3]
**Zhenghao Xing**[1], **Hongsheng Li**[2], **Pheng-Ann Heng**[1]

CUHK [1]IMIXR & [2]MMLab    [3]Peking University    [4]Shanghai AI Lab

[*]Equal Contribution    [†]Project Leader    [‡]Corresponding Author

## Abstract

Recent advancements underscore the significant role of Reinforcement Learning (RL) in enhancing the Chain-of-Thought (CoT) reasoning capabilities of large language models (LLMs). Two prominent RL algorithms, Direct Preference Optimization (DPO) and Group Relative Policy Optimization (GRPO), are central to these developments, showcasing different pros and cons. Autoregressive image generation, also interpretable as a sequential CoT reasoning process, presents unique challenges distinct from LLM-based CoT reasoning. These encompass ensuring text-image consistency, improving image aesthetic quality, and designing sophisticated reward models, rather than relying on simpler rule-based rewards. While recent efforts have extended RL to this domain, these explorations typically lack an in-depth analysis of the domain-specific challenges and the characteristics of different RL strategies. To bridge this gap, we provide the first comprehensive investigation of the GRPO and DPO algorithms in autoregressive image generation, evaluating their ***in-domain*** performance and ***out-of-domain*** generalization, while scrutinizing the impact of ***different reward models*** on their respective capabilities. Our findings reveal that GRPO and DPO exhibit distinct advantages, and crucially, that reward models possessing stronger intrinsic generalization capabilities potentially enhance the generalization potential of the applied RL algorithms. Furthermore, we systematically explore ***three prevalent scaling strategies*** to enhance both their in-domain and out-of-domain proficiency, deriving unique insights into efficiently scaling performance for each paradigm. We hope our study paves a new path for inspiring future work on developing more effective RL algorithms to achieve robust CoT reasoning in the realm of autoregressive image generation. Code is released at https://github.com/ZiyuGuo99/Image-Generation-CoT.

## 1 Introduction

Recent large language models (LLMs) [33, 34, 47, 59, 64] have demonstrated remarkable achievements in diverse challenging tasks, such as mathematical problem-solving [4, 19, 32] and code generation [8, 5, 22]. This is driven by their emergent and robust reasoning capabilities through extended Chain-of-Thoughts (CoT) [52, 24, 18, 65, 17, 66], as exemplified by models like OpenAI's o1 [35], DeepSeek-R1 [16], and Kimi k1.5 [45]. These predominant advancements in reasoning abilities are primarily facilitated by reinforcement learning (RL) [3, 61, 30] methods, applied during post-training, which elicits a deliberate, stepwise reasoning process before reaching a final answer.

Among the prominent RL algorithms employed for fine-tuning LLMs, two representative approaches stand out, i.e., Direct Preference Optimization (DPO) [39] and Group Relative Policy Optimization

39th Conference on Neural Information Processing Systems (NeurIPS 2025).

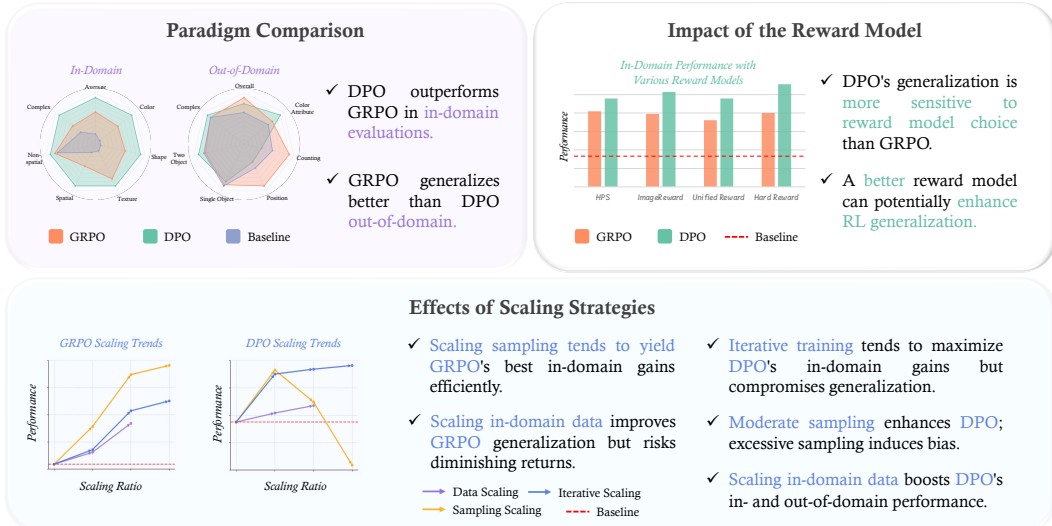

Figure 1: **Investigation for GRPO and DPO in Autoregressive Image Generation.** We analyze the advantages of GRPO and DPO in both in-domain and out-of-domain scenarios (Top-left), the effect of different reward models (Top-right), and the influence of scaling strategies (Bottom), providing unique insights to this field.

(GRPO) [43]. DPO offers compelling advantages in reduced training time and computational cost, while achieving substantial performance gains, especially in tasks with simpler and shorter CoT reasoning [37, 57]. Nevertheless, DPO's effectiveness diminishes in more complex reasoning scenarios due to its reliance on static, pre-collected data, making it increasingly prone to noise and biases as task complexity rises [57]. In contrast, recent studies underscore GRPO's superior ability to tackle complicated challenges requiring intricate CoT reasoning [16]. It achieves this by iteratively refining policies using self-generated data, effectively adapting to complex task distributions. Nevertheless, GRPO consistently incurs significantly higher computational costs and protracted training durations, as it samples the policy online and continuously during training.

In parallel, image generation [26, 12, 38, 40], one of the most fundamental tasks in multi-modality, also requires substantial reasoning knowledge. Notably, autoregressive generation models [44, 55, 9, 53] can be viewed as a form of CoT reasoning similar to that of LLMs, as they sequentially predict visual tokens in a step-by-step manner. One preliminary work, *'Image Generation with CoT'* [18] has verified the feasibility of DPO in this domain, and follow-up studies [23, 48, 58] experiment with GRPO. However, these endeavors lack in-depth investigation, since the CoT in visual generation exhibits several critical distinctions from textual reasoning. First, image generation tasks often involve prompts describing diverse scenarios, ranging from intricate, detailed descriptions to concise, templated formats. This frequently exhibits a substantial domain gap, with generative models typically excelling in one domain but may struggle in another. Consequently, evaluating the efficacy of RL from both in-domain and out-of-domain perspectives is crucial. Second, distinct from the often verifiable, rule-based rewards of LLMs, image generation objectives, which rely on diverse criteria like text-image alignment and human aesthetic preferences, necessitate a thorough exploration of how different reward models affect the performance of RL algorithms. Third, while recent works, such as [18], have preliminarily adopted scaling strategies like iterative-DPO [37] to improve performance, the impact of many widely used scaling strategies still remains insufficiently explored. Given these challenges, we raise the question: *How does the performance of GRPO compare to that of DPO in autoregressive image generation, and which aspects may influence their optimal performance, e.g., reward models and scaling strategies?*

To this end, we conduct a systematic investigation comparatively evaluating GRPO and DPO regarding their capacity for CoT reasoning in autoregressive image generation. We adopt Janus-Pro [9] as our baseline autoregressive generative model, evaluating its in-domain performance on T2I-CompBench [21] and out-of-domain generalization on GenEval [15]. Specifically, as illustrated in Figure 1, our investigation delves into three primary aspects concerning both GRPO and DPO:

- **In-Domain Performance *vs* Out-of-Domain Generalization.** Examining both in-domain and out-of-domain scenarios enables a comprehensive evaluation of a model's robustness in handling prompts of varying granularity. In our settings, we leverage preference data derived from a unified reward model, conducting a rigorous comparative analysis.

  *__Observation:__ 1) The off-policy DPO method demonstrates superior performance on in-domain tasks compared to GRPO. 2) Conversely, GRPO exhibits stronger generalization capabilities, outperforming DPO on the out-of-domain benchmark.*

- **Impact of Different Reward Models.** Reward models define the preference distribution, with diverse models [54, 56, 51, 18] serving this role in text-to-image generation. Understanding how the choice of reward model influences policy outcomes, particularly generalization, is crucial for preference-based methods. To this end, we investigate the differential impact of employing various reward models on the generalization performance of GRPO and DPO, revealing how preference variations shape algorithmic capabilities.

  *__Observation:__ 1) DPO exhibits greater sensitivity to reward model variations than GRPO, manifesting larger out-of-domain performance fluctuations. 2) A reward model with superior generalization can potentially improve the generalization performance of RL algorithms.*

- **Investigation of Effective Scaling Strategies.** Investigating how prevalent scaling strategies affect RL's in-domain and out-of-domain performance offers key insights into enhancing model adaptability across tasks. We examine three dimensions: scaling sampled images per prompt, scaling in-domain training data diversity and volume, and adopting an iterative training approach inspired by Iterative-DPO.

  *__Observation:__ GRPO: 1) Scaling sampled images tends to yield more computationally efficient in-domain gains. 2) Moderate scaling of sampling size and in-domain data improve generalization, but excessive scaling risks overfitting. DPO: 1) Scaling iterative training tends to maximize in-domain performance but compromise generalization after multiple cycles. 2) Moderate sampling sharpens preference contrast, optimizing both in-domain and out-of-domain performance, while excessive sampling induces bias. 3) Scaling in-domain data enhances both in-domain and generalization performance by mitigating biases introduced by the limited preference scope of small datasets.*

In summary, our core contributions are as follows:

- We present *the first* comprehensive empirical study comparing GRPO and DPO algorithms for autoregressive image generation, highlighting their respective strengths and providing unique insights into the future advancement of this field.

- By evaluating the inherent generalization capabilities of various reward models and their influence on both GRPO and DPO, we demonstrate that enhancing the generalization capacity of reward models potentially boosts the overall generalization performance of RL.

- We systematically investigate scaling behaviors across multiple critical dimensions, including variations in the number of sampled images per prompt, the scale of the in-domain training data, and the deployment of iterative training paradigms, with the analysis of their differential impact on in-domain performance and out-of-domain generalization, yielding numerous valuable insights and highlighting promising avenues for future research.

## 2   Our Investigation

Group Relative Policy Optimization (GRPO) [43] and Direct Preference Optimization (DPO) [39] have demonstrated distinct advantages for fine-tuning large language models (LLMs). In this study, following the step of *'Image Generation with CoT'* [18], we conduct a meticulous and systematic investigation aiming to evaluate the efficacy of GRPO and DPO in image generation and identify more impactful strategies for scaling performance.

Table 1: **In-Domain Performance of GRPO and DPO under Various Reward Models.** We assess the performance on the T2I-CompBench benchmark [21], comprehensively evaluating different reward models including HPS [54], ImageReward [56], Unified Reward [51], Fine-tuned ORM [18], and 'Metric Reward'.

| Reward Type | Average | | Attribute Binding | | | | | | Object Relationship | | | | Complex | |
|---|---|---|---|---|---|---|---|---|---|---|---|---|---|---|
| | | | Color | | Shape | | Texture | | Spatial | | Non-Spatial | | | |
| | GRPO | DPO | GRPO | DPO | GRPO | DPO | GRPO | DPO | GRPO | DPO | GRPO | DPO | GRPO | DPO |
| Baseline | 38.56 | | 63.30 | | 34.28 | | 48.90 | | 20.23 | | 30.51 | | 34.12 | |
| HPS | 50.49 | **53.90** | 77.39 | 85.25 | 53.59 | 64.72 | 71.54 | 76.08 | 30.14 | 25.29 | 31.10 | 31.17 | 39.20 | 40.89 |
| ImageRwd | 49.76 | **55.67** | 76.57 | 86.18 | 52.58 | 64.92 | 70.71 | 76.20 | 28.82 | 34.72 | 31.10 | 31.53 | 38.80 | 40.47 |
| Unified Rwd | 48.01 | **53.91** | 74.34 | 82.88 | 50.28 | 61.74 | 68.66 | 76.64 | 27.80 | 30.40 | 30.87 | 31.25 | 36.10 | 40.52 |
| Ft. ORM | 53.11 | **55.10** | 79.49 | 84.92 | 56.96 | 60.87 | 75.58 | 76.76 | 35.97 | 35.97 | 31.10 | 31.26 | 39.61 | 40.79 |
| Metric Rwd | 50.01 | **57.81** | 77.21 | 87.83 | 53.66 | 65.91 | 73.17 | 79.19 | 28.69 | 40.76 | 30.87 | 31.53 | 36.97 | 41.64 |

Table 2: **Out-of-Domain Performance of GRPO and DPO under Various Reward Models.** We assessed the generalization performance of both algorithms on the out-of-domain GenEval dataset [15], with reward models aligning consistently with Table 1.

| Reward Type | Overall | | Color Attr | | Counting | | Position | | Single Object | | Two Object | | Colors | |
|---|---|---|---|---|---|---|---|---|---|---|---|---|---|---|
| | GRPO | DPO | GRPO | DPO | GRPO | DPO | GRPO | DPO | GRPO | DPO | GRPO | DPO | GRPO | DPO |
| Baseline | 78.04 | | 63.50 | | 54.37 | | 76.25 | | 98.44 | | 87.63 | | 88.03 | |
| HPS | **79.18** | 77.31 | 63.00 | 70.25 | 60.62 | 48.12 | 78.75 | 67.25 | 99.69 | 90.15 | 86.87 | 98.75 | 86.17 | 89.36 |
| ImageRwd | **79.26** | 77.67 | 59.50 | 69.25 | 62.50 | 43.44 | 80.50 | 71.25 | 98.44 | 99.69 | 87.37 | 90.66 | 87.23 | 91.76 |
| Unified Rwd | **80.99** | 80.03 | 67.52 | 70.54 | 61.46 | 55.69 | 82.02 | 75.24 | 99.58 | 99.00 | 88.40 | 91.50 | 86.96 | 88.21 |
| Metric Rwd | **79.44** | 78.86 | 65.75 | 70.29 | 58.13 | 51.98 | 79.50 | 75.24 | 98.12 | 97.46 | 87.12 | 89.47 | 88.03 | 88.73 |

## 2.1 Overview

**Task Formulation.** To enable the applicability of the prevailing two RL techniques, namely GRPO and DPO, we focus on the autoregressive image generation task, demonstrated by models such as LlamaGen [44], Show-o [55], and Janus-Pro [9]. These models employ a data representation and output paradigm analogous to that of LLMs and large multimodal models (LMMs), while attaining comparable performance to continuous diffusion models [20]. Specifically, they leverage quantized autoencoders [13] to transform images into discrete tokens, enabling the seamless integration of loss mechanisms from both DPO and GRPO during the post-training phase.

**Experimental Settings.** We select Janus-Pro as our baseline model for this investigation, a latest autoregressive image generation model with advanced capabilities. To comprehensively evaluate the effectiveness of various RL strategies, we assess the text-to-image generation performance on both in-domain and out-of-domain benchmarks. In-domain performance is evaluated using T2I-CompBench [21], which features long, detailed prompts designed to compose complex scenes with multiple objects, displaying various attributes and relationships. Out-of-domain generalization is examined with GenEval [15], utilizing short, templated prompts starting with "a photo of," to assess robustness to concise and standardized textual prompts. In the subsequent sections, we explore three pivotal dimensions to investigate GRPO and DPO for implementing chain-of-thought reasoning in autoregressive image generation: in-domain *vs.* out-of-domain performance (Section 2.2), the impact of reward models (Section 2.3), and the effectiveness of scaling strategies (Section 2.4).

## 2.2 In-Domain Performance *vs* Out-of-Domain Generalization

Recently, the application of reinforcement learning (RL) has yielded substantial breakthroughs in lifting the reasoning capabilities of LLMs via Chain-of-Thought (CoT) techniques [52, 35, 11, 45, 2, 1]. Existing researches indicate that GRPO, the on-policy RL algorithm, can data-efficiently enhance performance on in-domain (ID) tasks without sacrificing capabilities on out-of-domain (OOD) tasks [28, 31, 14, 29]. Nevertheless, a direct comparison of on-policy GRPO and off-policy Deep Policy Optimization (DPO) regarding their ID and OOD performance under equivalent training data remains underexplored. Drawing inspiration from this, we train and evaluate these two algorithms within the domain of autoregressive image generation to elucidate their comparative strengths.

**GRPO.** GRPO improves upon Proximal Policy Optimization (PPO) by eliminating the learned value critic and estimating advantages directly through group-wise normalization of rewards across

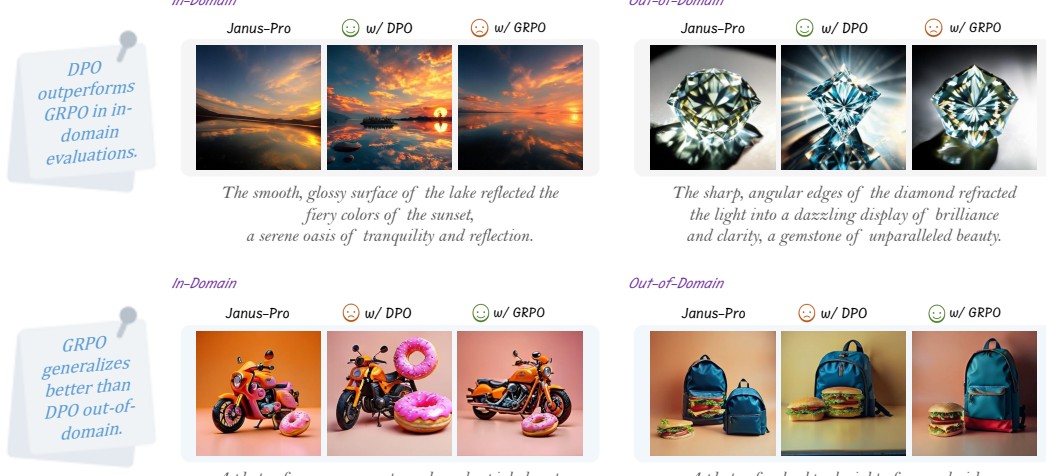

Figure 2: **Visualization Results of In-Domain *vs*. Out-of-Domain Performance Comparison.**

multiple responses per query. Our empirical investigation initiates with a GRPO training phase, employing a group size of four completions per query with the hyperparameter for iteration times set to 1, following the standard GRPO workflow. Building on the design of reward models, as elaborated in 2.3, we focus here on the curation of training data for GRPO. To facilitate a fair comparison between GRPO and DPO, we ensured both methods utilized identical training data. As GRPO's on-policy framework eliminates the need for auxiliary dataset construction, we trained models using the official prompts from the T2I-CompBench dataset, totaling 5.6k prompts.

**DPO.** In contrast to the GRPO on-policy RL paradigm, which solely requires training prompts, DPO functions as an off-policy algorithm, necessitating both prompts and corresponding pairs of chosen and rejected images. Given that autoregressive image generation models are optimized using a cross-entropy loss, we can directly adapt the maximum likelihood objective of DPO to this setting. To clarify, we provide a detailed outline of the DPO methodology from the following two perspectives:

- *Maintaining Comparable Computational Cost:* The computational cost of DPO comprises three components: (i) generating training images based on provided prompts, (ii) the scoring of these images by a reward model, and (iii) the subsequent training process. To ensure a fair comparison between DPO and GRPO under comparable computational constraints, we align the number of images generated per prompt in DPO with the group size in GRPO, while employing identical reward models to maintain consistent learning preferences.

- *DPO Ranking Data Curation:* Leveraging the images generated by the model, we construct ranking pairs for each prompt by selecting the highest and lowest scoring images as the chosen and rejected images, respectively. This process yields a total of 5.6k ranking pairs for training, derived from the official prompts of the T2I-CompBench dataset.

**Experimental Analysis and Insights.** As presented in Table 1 and 2, we evaluate the in-domain and out-of-domain performance of GRPO and DPO. We provide qualitative results in Figure 2, with additional visualization in the Supplementary. The key findings are summarized as follows:

- *DPO demonstrates stronger performance than GRPO in in-domain evaluations.* As shown in Table 1, DPO consistently outperforms GRPO on T2I-CompBench across various reward models, with DPO's in-domain performance surpassing GRPO by an average of 11.53%. Notably, when using T2I-CompBench's official evaluation tools as the reward signal, DPO attains a peak enhancement of 7.8% over GRPO. This significantly highlights the advantages of DPO over GRPO in terms of in-domain effectiveness and robustness.

- *GRPO exhibits superior generalization capabilities than DPO in out-of-domain scenarios.* As illustrated in Table 2, GRPO consistently demonstrates enhanced generalization performance over DPO across various reward models on the GenEval dataset, surpassing DPO by

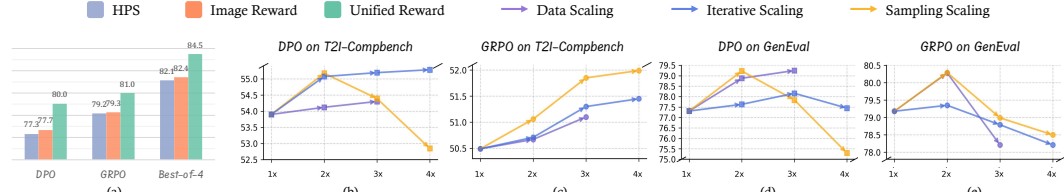

Figure 3: **(a) The Impact of Different Reward Models' Intrinsic Generalization Capability.** We evaluate the generalization performance of GRPO, DPO, and the intrinsic generalization performance (represented by best-of-4 strategy) of three reward models. **(b-e) Effects of Three Scaling Strategies.** We examine the effects of various scaling strategies, including sampling size, in-domain data diversity, and iterative training, on both in-domain and out-of-domain performance.

an average of 1.14%. Notably, when HPS serves as the reward model, GRPO achieves a peak improvement of 2.42% over DPO, suggesting its superior generalization capacity.

## 2.3 Impact of Different Reward Models

Recent advancements in LLMs have extensively investigated the influence of reward model variations on performance across diverse tasks, including reasoning [52, 50], safety [36, 67], and general conversation [46, 49]. In text-to-image generation, reward models have been developed to capture human aesthetic and semantic preferences, thereby steering the generative process. However, in autoregressive image generation, the impact of reward model-induced preferences on RL remains limited. A recent study [23] explores the influence of reward models on in-domain RL performance but still lacks exploration of the potential relationship between the intrinsic properties, especially the generalization capabilities, of reward models and RL generalization. To address this gap, we consider exploring the relationship between the generalization of RL and the intrinsic generalization capabilities of reward models, yielding critical insights for the future development of reward models.

**Reward Model Design.** Unlike domains such as mathematics or programming, where reward signals are typically derived from verification functions that ensure precise alignment with ground-truth solutions, evaluating the quality of generated images necessitates sophisticated learned reward models. The influence of these reward models, along with their inherent biases, on the training process remains underexplored. To elucidate their impact, we examine three distinct types of reward models, each designed to capture unique aspects of image quality and alignment with human preferences:

- **Human Preference Model.** Human Preference Models, such as HPS [54] and ImageReward [56], are constructed using vision-language models (VLMs) like CLIP or BLIP to evaluate images based on human aesthetic appeal and text-image alignment. Trained on datasets of human-annotated image rankings, these models provide a holistic assessment of visual quality, generating scores that reflect human-like preferences.

- **Visual Question Answering Model.** Visual Question Answering Models, including UnifiedReward [51], Fine-tuned ORM [18] and PARM [18], leverage multimodal large language models (MLLMs) like LLaVA [27] to interpret visual inputs and perform scenario-based evaluations. Trained on diverse datasets comprising images and text, these models emphasize detailed reasoning, enabling precise scoring and evaluation of visual content.

- **Metric Reward.** Metric Rewards utilize specialized, domain-specific evaluation tools. In this study, for each training prompt, we identify its corresponding attribute in T2I-CompBench and apply the associated evaluation protocol to score the generated images.

**Intrinsic Generalization of Reward Models.** In contrast to RL, we adopt Guo et al.'s scalable framework ([18]) for efficiently evaluating reward model (RM) capabilities during inference. This approach uses RMs as outcome reward models (ORMs) in a best-of-N strategy, assessing their capabilities via final scores. We extend this framework to evaluate RM generalization on the GenEval dataset by deploying RMs as ORMs. As shown in Table 3, a best-of-4 selection strategy yields the RM generalization ranking on GenEval: **Unified Reward > Image Reward > HPS Reward**.

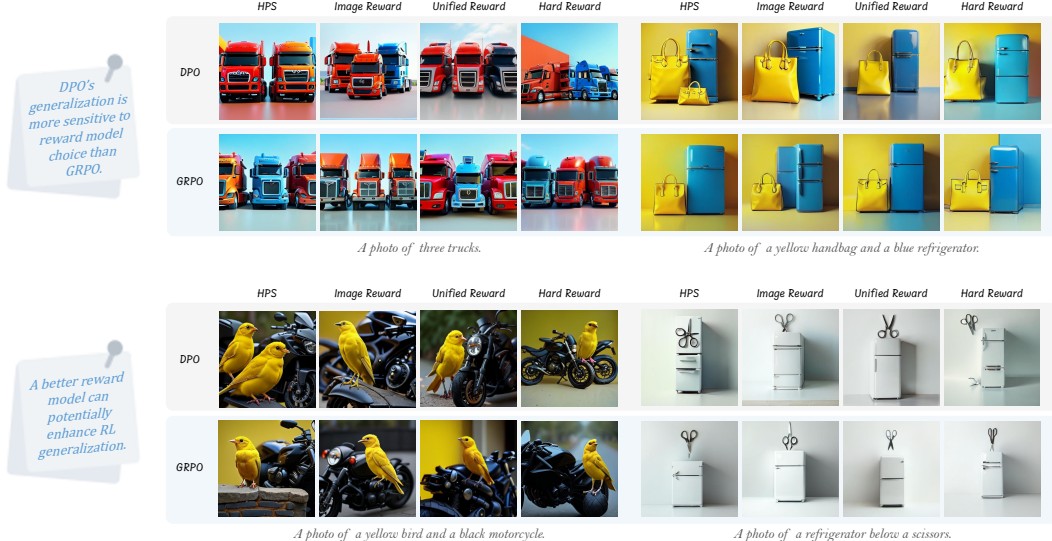

Figure 4: **Visualization Results of the Impact of Different Reward Models.**

Table 3: **Comparison of Reward Model Generalization.** We assess the generalization capabilities of reward models in a best-of-4 selection strategy on the GenEval dataset [15].

| Reward Type | Overall | Color Attr. | Counting | Position | Single Obj. | Two Obj. | Colors |
|---|---|---|---|---|---|---|---|
| HPS | 82.14 | 70.62 | 77.25 | 90.15 | 99.06 | 87.77 | 68.00 |
| ImageRwd | 82.41 | 69.69 | 75.50 | 93.18 | 98.75 | 88.56 | 68.75 |
| Unified Rwd | 84.49 | 75.00 | 82.00 | 92.42 | 98.75 | 87.50 | 71.25 |

**Experimental Analysis and Insights.** Our experimental results are visually summarized in Figure 3, and the detailed numeric comparisons are reported in Table 3. We provide qualitative results in Figure 4, with additional visualization in the Supplementary. Then, we draw two principal insights:

- ***DPO's generalization performance exhibits heightened sensitivity to the choice of Reward Model compared to GRPO.*** As presented in Table 2, the range of GRPO's generalization performance on the GenEval dataset is 1.81, whereas DPO's range is notably higher at 2.72. Furthermore, GRPO's performance variance on GenEval is 0.5486, significantly lower than DPO's variance of 0.9547. This suggests that DPO's generalization capabilities are more susceptible to variations in training data preferences, indicating a greater dependency on the specific characteristics of the chosen reward model.

- ***A reward model with superior generalization can potentially improve the generalization performance of RL algorithms.*** As intuitively illustrated in Figure 3 (a), the performance rankings of different reward models on the GenEval dataset, optimized using either GRPO or DPO, remain consistent. Crucially, these rankings align perfectly with our prior evaluations of the intrinsic generalization capabilities of these models. This indicates that the reward model's intrinsic capacity for generalization is a pivotal factor that probably contributes to the overall generalization potential of the RL algorithm.

## 2.4 Investigation of Effective Scaling Strategies

Prior research has extensively explored methods to optimize the in-domain performance of RL algorithms for LLMs. Notable approaches include iterative DPO (DPO-Iter) [37] and techniques to enhance the efficacy of proximal policy optimization [57]. However, despite these advancements, common scaling behaviors to improve the in-domain (ID) and out-of-domain (OOD) performance of both on-policy and off-policy RL algorithms in autoregressive image generation remain largely underexplored. To address this gap, this section investigates three critical scaling behaviors to enhance the performance of GRPO and DPO across ID and OOD datasets. Specifically, these factors include:

Table 4: **Effect of Scaling Strategies on In-Domain Proficiency.** This table presents the performance evaluation of GRPO and DPO on T2I-CompBench [21] under three distinct scaling strategies: sample scaling (scaling sampled images per prompt), data scaling, and iterative training. Specifically, configurations denoted as 'Base Size' (Data Scaling), 'Sampling 4' (Sample Scaling), and 'Base' (Iterative Training) correspond to the values in Table 1 where the reward type is HPS for comparative analysis. All experiments consistently employ HPS as the reward model.

| Scaling Param. | Average | | Attribute Binding | | | | | | Object Relationship | | | | Complex | |
|---|---|---|---|---|---|---|---|---|---|---|---|---|---|---|
| | | | Color | | Shape | | Texture | | Spatial | | Non-Spatial | | | |
| | GRPO | DPO | GRPO | DPO | GRPO | DPO | GRPO | DPO | GRPO | DPO | GRPO | DPO | GRPO | DPO |
| Baseline | 38.56 | | 63.30 | | 34.28 | | 48.90 | | 20.23 | | 30.51 | | 34.12 | |
| *Data Scaling* | | | | | | | | | | | | | | |
| Base Size | 50.49 | 53.90 | 77.39 | 85.25 | 53.59 | 64.72 | 71.54 | 76.08 | 30.14 | 25.29 | 31.10 | 31.17 | 39.20 | 40.89 |
| Double Size | 50.67 | 54.12 | 77.81 | 82.24 | 54.39 | 60.48 | 72.27 | 77.06 | 29.16 | 33.05 | 31.15 | 31.18 | 39.26 | 40.70 |
| Triple Size | 51.11 | 54.30 | 77.73 | 83.77 | 56.31 | 62.35 | 72.66 | 77.73 | 29.21 | 29.81 | 31.18 | 31.28 | 39.54 | 40.84 |
| *Sample Scaling* | | | | | | | | | | | | | | |
| Sampling 4 | 50.49 | 53.90 | 77.39 | 85.25 | 53.59 | 64.72 | 71.54 | 76.08 | 30.14 | 25.29 | 31.10 | 31.17 | 39.20 | 40.89 |
| Sampling 8 | 51.06 | 55.17 | 77.79 | 83.31 | 55.47 | 63.86 | 72.63 | 75.50 | 29.99 | 36.44 | 31.19 | 31.05 | 39.30 | 40.84 |
| Sampling 12 | 51.85 | 54.39 | 78.18 | 82.37 | 56.81 | 63.17 | 73.72 | 77.82 | 31.51 | 31.57 | 31.21 | 31.14 | 39.68 | 40.28 |
| Sampling 16 | 51.99 | 52.85 | 77.38 | 80.60 | 59.09 | 63.49 | 73.72 | 75.72 | 30.70 | 25.93 | 31.15 | 31.06 | 39.88 | 40.31 |
| *Iterative Training* | | | | | | | | | | | | | | |
| Base | 50.49 | 53.90 | 77.39 | 85.25 | 53.59 | 64.72 | 71.54 | 76.08 | 30.14 | 25.29 | 31.10 | 31.17 | 39.20 | 40.89 |
| Iter1 | 50.71 | 55.07 | 77.28 | 85.14 | 54.14 | 64.21 | 72.35 | 76.15 | 29.78 | 34.05 | 31.18 | 31.28 | 39.51 | 39.58 |
| Iter2 | 51.30 | 55.19 | 78.02 | 84.66 | 55.73 | 64.44 | 72.86 | 76.15 | 30.76 | 35.05 | 31.16 | 31.29 | 39.25 | 39.54 |
| Iter3 | 51.45 | 55.28 | 77.97 | 84.70 | 55.94 | 64.32 | 73.26 | 76.32 | 30.74 | 35.51 | 31.19 | 31.27 | 39.61 | 39.53 |

Table 5: **Effect of Scaling Strategies on Out-of-Domain Generalization.** This evaluation on the GenEval [15] dataset adopts the same scaling strategies as outlined in T2I-CompBench (see Table 4). The configurations, Base Size (Data Scaling), Sampling 4 (Sample Scaling), and Base (Iterative Training), correspond to the values in Table 2, where the reward type is HPS for comparison.

| Scaling Param. | Overall | | Color Attr | | Counting | | Position | | Single Object | | Two Object | | Colors | |
|---|---|---|---|---|---|---|---|---|---|---|---|---|---|---|
| | GRPO | DPO | GRPO | DPO | GRPO | DPO | GRPO | DPO | GRPO | DPO | GRPO | DPO | GRPO | DPO |
| Baseline | 78.04 | | 63.50 | | 54.37 | | 76.25 | | 98.44 | | 87.63 | | 88.03 | |
| *Data Scaling* | | | | | | | | | | | | | | |
| Base Size | 79.18 | 77.31 | 63.00 | 70.25 | 60.62 | 48.12 | 78.75 | 67.25 | 86.87 | 90.15 | 99.69 | 98.75 | 86.17 | 89.63 |
| Double Size | 80.28 | 78.88 | 66.50 | 67.50 | 62.81 | 50.31 | 78.25 | 77.00 | 87.88 | 91.67 | 98.75 | 99.06 | 87.50 | 87.88 |
| Triple Size | 78.21 | 79.25 | 64.50 | 70.00 | 56.56 | 51.25 | 73.75 | 76.00 | 87.12 | 91.67 | 98.75 | 99.06 | 88.56 | 87.50 |
| *Sample Scaling* | | | | | | | | | | | | | | |
| Sampling 4 | 79.18 | 77.31 | 63.00 | 70.25 | 60.62 | 48.12 | 78.75 | 67.25 | 86.87 | 90.15 | 99.69 | 98.75 | 86.17 | 89.63 |
| Sampling 8 | 80.29 | 79.23 | 63.25 | 73.25 | 63.44 | 47.81 | 80.25 | 70.00 | 90.15 | 95.45 | 96.88 | 98.75 | 87.77 | 90.16 |
| Sampling 12 | 78.99 | 77.84 | 65.25 | 66.25 | 60.62 | 47.81 | 75.50 | 70.25 | 89.14 | 92.68 | 97.81 | 99.38 | 85.64 | 90.69 |
| Sampling 16 | 78.50 | 75.30 | 63.50 | 64.75 | 57.81 | 44.69 | 76.75 | 63.00 | 86.62 | 91.16 | 99.38 | 99.38 | 86.97 | 88.83 |
| *Iterative Training* | | | | | | | | | | | | | | |
| Base | 79.18 | 77.31 | 63.00 | 70.25 | 60.62 | 48.12 | 78.75 | 67.25 | 86.87 | 90.15 | 99.69 | 98.75 | 86.97 | 89.36 |
| Iter1 | 79.35 | 77.63 | 67.00 | 69.00 | 58.75 | 54.37 | 76.50 | 69.00 | 87.63 | 90.91 | 98.75 | 98.75 | 87.50 | 83.78 |
| Iter2 | 78.79 | 78.16 | 64.75 | 71.00 | 57.19 | 56.62 | 76.00 | 66.50 | 89.14 | 92.68 | 98.44 | 97.81 | 87.23 | 85.37 |
| Iter3 | 78.21 | 77.45 | 61.36 | 65.50 | 60.63 | 57.81 | 75.75 | 68.75 | 87.88 | 91.16 | 97.50 | 97.19 | 86.44 | 84.31 |

(1) scaling sampled images per prompt, (2) scaling the diversity of in-domain training data, and (3) implementing iterative training paradigms. The detailed application is delineated as follows:

- *Scaling Sampled Images per Prompt:* For GRPO, scaling the quantity of sampled images per prompt corresponds to expanding the group size of real-time samples utilized during the RL training phase. For DPO, where pairs are constructed by selecting the highest- and lowest-scoring images from a pre-generated set for a given prompt, this strategy effectively amplifies the discriminative power of the preference pairs, facilitating more precise alignment with human preferences and the model's ability to distinguish subtle quality differences.

- *Scaling the Diversity and Quantity of In-Domain Training Data:* To scale data while rigorously maintaining quality control, we develop a structured prompt generation pipeline leveraging GPT-4o. Building upon the T2I-CompBench, we generated a set of category-specific prompts that was twice as large through carefully constrained API calls (see Supplementary for more implementation details). Our methodology incorporated two key principles:
  1. *Category-Aware Constraint Preservation:* All generated prompts maintained strict adherence to their respective category's syntactic templates and semantic boundaries (e.g., 3D spatial prompts required exactly two objects and one spatial relation).

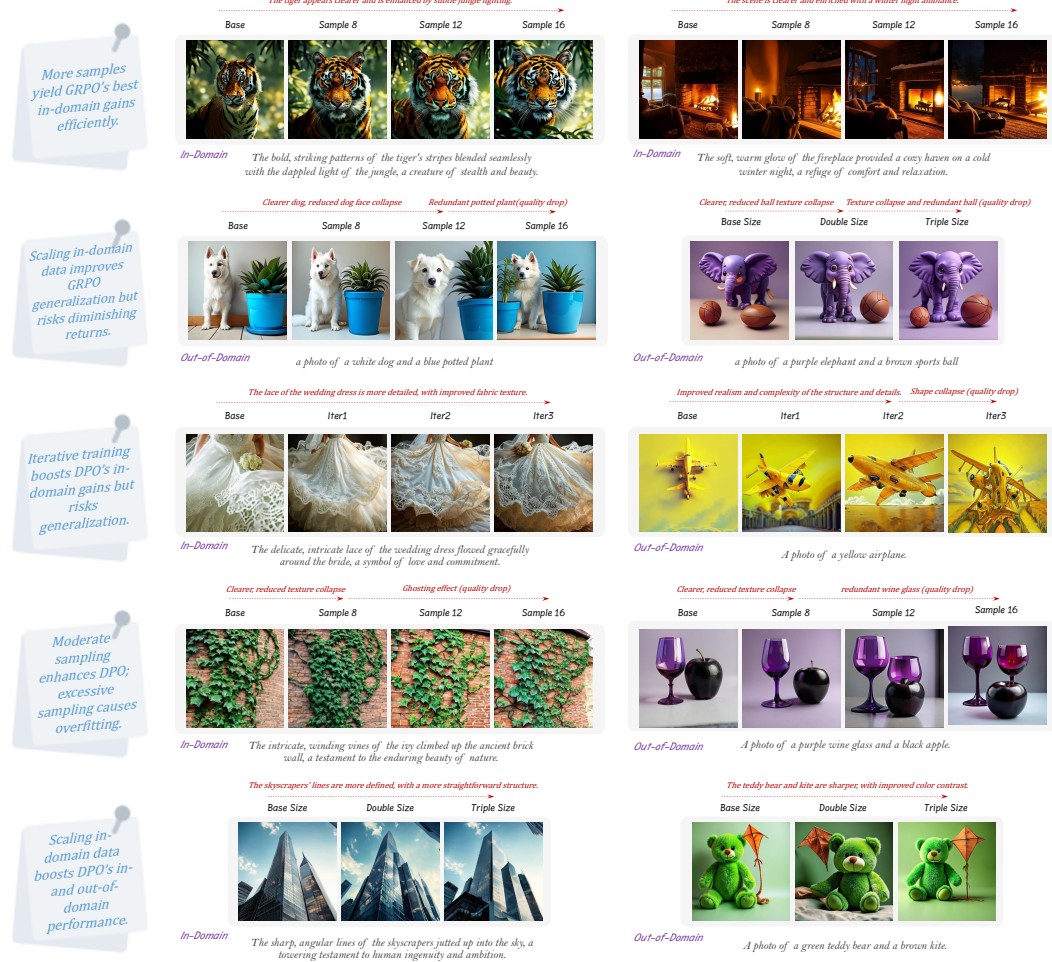

Figure 5: **Visualization Results of Insights from Investigating Scaling Strategies.**

2. ***Semantic Novelty Enforcement:*** We implemented specific generation constraints to prevent superficial variations, requiring GPT-4o to produce genuinely novel compositions rather than simple lexical substitutions. This yields semantically distinct yet plausible object pairings (e.g., *"A person is exploring a forest and taking photos of the wildlife"* ***vs.*** original *"A child is playing with a toy airplane in the backyard"*), significantly expanding the conceptual coverage of our training data.

- ***Implementing Iterative Training Paradigms:*** The motivation for adopting iterative training paradigms lies in their ability to progressively enhance model performance by leveraging updated reference policies, thereby reducing overfitting risks and improving generalization. Inspired by this, we develop iterative variants of GRPO and DPO, termed GRPO-Iter and DPO-Iter, respectively. For GRPO-Iter, we extend the standard GRPO framework with iterative cycles, updating the reference model with the policy model's parameters after each training round, which ensures the KL penalty aligns with the current policy. For DPO-Iter, we followed iterative DPO protocols[37, 57], where preference pairs are re-sampled and re-evaluated after each training cycle, with the reference model updated accordingly.

**Experimental Analysis and Insights.**    As shown in Table 4 and 5, we conduct a comprehensive investigation into the efficacy of three key scaling strategies. To enable a fair comparison of GRPO and DPO, we standardize the experimental setting by adopting the HPS [54] as the reward model to compare the effectiveness of the three scaling strategies, with additional visualizations provided in the Supplementary due to space limitations. Our principal findings are summarized as follows:

- ***Scaling sampled images per prompt tends to yield more computationally efficient in-domain gains for GRPO:*** Qualitative visualizations in Figure 3 (c) for GRPO on T2I-CompBench demonstrate that scaling group size in autoregressive image generation outperforms both in-domain data scaling and iterative scaling. This enhanced performance is driven by improved advantage estimation with larger group sizes, which stabilizes policy updates and strengthens in-domain optimization. Given that doubling each factor (group size, in-domain data scale, or iterative training paradigms) incurs approximately comparable computational cost, prioritizing scaling sampling emerges as a highly efficient strategy.

- ***Moderately scaling the sample size and increasing the diversity and quantity of in-domain data improves generalization for GRPO, but excessive scaling diminishes growth:*** In autoregressive image generation, moderate sampling size within the GRPO algorithm (e.g., to 4 or 8) and in-domain training data quantity (e.g., by doubling or tripling) progressively improves generalization, as shown in Figure 3 (e) for GRPO on GenEval. However, when larger sample sizes are employed or when GRPO's in-domain data scale is *triple*, generalization exhibits a slight decline due to overfitting to in-domain characteristics, which underscores the critical need to balance in-domain optimization with robust generalization.

- ***Iterative training tends to maximize DPO's in-domain performance but risks generalization degradation:*** Iterative training substantially enhances the in-domain performance of DPO for autoregressive image generation, as evidenced by the Iterative Scaling curve in Figure 3 (b) and (d). Notably, a single iteration of DPO-Iter outperforms the results obtained by tripling the training data, with additional iterations providing further incremental gains in in-domain metrics. However, generalization on the GenEval declines significantly after two iterations, likely due to overfitting the training preference data, underscoring the trade-off between maximizing preference alignment and maintaining robust generalization.

- ***Moderate sampling optimizes DPO's preference contrast for improved in-domain and generalization performance, while excessive sampling induces bias:*** In autoregressive image generation, scaling the sample size for DPO preference pair selection yields non-monotonic performance effects, as illustrated by the DPO sampling scaling curve in Figure 3 (b) and (d). Relative to the baseline, sample sizes of 4, 8, and 16 yield in-domain gains of 37.38%, 40.61%, and 34.71%, alongside out-of-domain generalization changes of $-0.93\%$, $+1.52\%$, and $-3.51\%$, respectively. This suggests that scaling the sample size optimizes preference contrast while avoiding biases introduced by excessive scaling.

- ***Scaling in-domain data for DPO optimizes performance across both in-domain and out-of-domain by mitigating preference bias:*** Scaling in-domain data by factors of one, two, and three achieves relative in-domain improvements of 37.40%, 37.97%, and 38.37% over the baseline, respectively. While single-scale training results in a 0.94% decline on the out-of-domain benchmark, scaling by factors of two and three produces gains of 1.08% and 1.55%. This highlights that carefully curating a diverse and representative set of preference pairs is critical to overcoming the constrained preference scope inherent in small datasets, thereby mitigating potential in domain and out-of-domain performance degradation.

## 3   Conclusion

In this paper, we conducted a rigorous experimental analysis demonstrating that DPO excels in in-domain tasks, while GRPO exhibits superior out-of-domain generalization. We further establish that the generalization capacity of reward models potentially shapes both algorithms' generalization potential. Through systematic exploration of three scaling strategies, we derive critical insights for achieving enhanced Chain-of-Thought reasoning in autoregressive image generation.

## Acknowledgement

The work described in this paper was supported in part by the Research Grants Council of the Hong Kong Special Administrative Region, China, under Project 14201321 and Project 14200824.

## Overview of Appendix

- Appendix A: Related work.

- Appendix B: Implementation details of structured prompt generation pipeline.

- Appendix C: Detailed record of computational time.

## A  Related Work

**Visual Generative Models.**  Visual generative models have advanced through two primary paradigms: autoregressive and diffusion approaches. Autoregressive methods, inspired by language modeling success [33, 34, 47, 59], sequentially predict image tokens or pixels, as seen in ViT-VQGAN [60] and VideoPoet [25]. Recent work like LlamaGen [44] demonstrates that pure autoregressive architectures can achieve state-of-the-art generation, while Janus [9] introduces decoupled visual encoding to unify multimodal understanding and generation. Meanwhile, diffusion models have emerged as a powerful alternative, with continuous approaches [63] dominating text-to-image tasks and discrete variants like MaskGIT [7] operating on tokenized representations. Notably, Show-o adopts discrete diffusion through masked token prediction, achieving high-fidelity generation while maintaining training efficiency.

**Reinforcement Learning (RL).**  Reinforcement Learning (RL) trains agents to maximize rewards through environment interactions, with methods split into on-policy (e.g., PPO [42], GRPO [43]) and off-policy (e.g., DPO [39]) approaches. On-policy methods like PPO use current policy data for stable but costly updates, employing techniques like GAE [41] for variance reduction, while GRPO replaces critics with group-wise reward comparisons. Off-policy methods like DPO reuse historical data for efficiency but risk distribution mismatch, directly optimizing preferences without reward modeling. The key difference lies in data usage: on-policy requires fresh data for stability, whereas off-policy trades some reliability for sample efficiency. Applied to language models via RLHF [6], these methods enhance alignment (e.g., RLOO [3]'s critic-free approach) and reasoning [10, 16, 45, 62] through MDP formulations, balancing computational cost and performance in tasks like mathematical reasoning. This demonstrates RL's adaptability across policy paradigms for improving language models.

## B  Implementation Details of Structured Prompt Generation Pipeline

As discussed in Sec. 2.4 of the main paper (*Investigation of Effective Scaling Strategies*), we enlarge T2I-COMPBENCH by generating an additional set of category-specific prompts with GPT-4o, thereby **doubling** the size of the original benchmark. Specifically, for each of the eight categories, *color*, *texture*, *shape*, *numeracy*, *spatial*, *3D spatial*, *non-spatial*, and *complex*, we craft a dedicated *meta-prompt*. All meta-prompts are derived from a shared template, but include category-dependent constraints. An example for the *color* category is given after the following paragraph.

In practice, we iterate through every prompt in the *color* subset of T2I-COMPBENCH, replace the placeholder #Prompts From T2I-CompBench# with the current prompt, and feed the resulting meta-prompt to GPT-4o. We apply the same pipeline to the remaining seven categories. The complete collection of category-specific meta-prompts will be released upon the paper's acceptance.

```
I am working on a reinforcement learning for image generation project, and I
need your assistance in generating additional prompts that focus on color-based
descriptions.
Existing prompts:  #Prompts From T2I-CompBench#
Task:  Generate 2 additional prompts that maintain the same syntactic structure
while ensuring diversity.
Requirements:
Color Usage:
Each prompt must explicitly include at least two different colors.
The color words should be commonly used and perceptually distinct (e.g., "red"
and "blue" are good, but "light red" and "dark red" are too similar).
Allowed color descriptors:  basic colors (e.g., red, blue, green, yellow, pink,
purple, orange, brown, black, white, gray) and common material-based variations
(e.g., "golden", "silver", "ivory").
Avoid uncommon or overly specific colors (e.g., "cerulean", "chartreuse").
Object Selection:
The first object should be a tangible item with a strong association to color
(e.g., clothing, furniture, makeup, vehicles, buildings).
The second object (if applicable) should also be a realistic, color-relevant
entity that fits within a scene.
Avoid repetition of objects already in the dataset (e.g., if "lipstick" and
"blush" exist, do not use them again).
Color and Object Compatibility:
Ensure that the selected colors are realistically applicable to the given
objects.
Examples of Good Color-object Pairings:
"A red sports car and a black leather seat." (both colors are reasonable for cars
and seats)
Examples of Bad Color-object Pairings (to avoid):
"A purple banana and a silver cloud." (unnatural color choices)
Diversity Constraints:
Do not generate prompts that are simple color swaps (e.g., "A red lipstick and a
pink blush" and "a pink lipstick and a red blush" are too similar).
Ensure semantic diversity by describing different types of objects and settings
(e.g., fashion, interior design, nature, technology).
The sentence structure should mimic the provided examples but not be identical.
Output Format:
Return the response as a Python list of strings in JSON-compatible format, e.g.:
{
"prompt1",
"prompt2"
}
Strictly use lowercase (no capitalization except for proper nouns).
Now, generate two new prompts following these requirements.
```

Table 6: **Comparison of DPO and GRPO Training Computational Costs (in GPU hours).**

| Reward Type | DPO | | | | GRPO |
| --- | --- | --- | --- | --- | --- |
| | *Simple Image* | *Scoring* | *Training* | *Total* | *Total* |
| HPS | 1.51 h | 0.83 h | 0.67 h | 2.99 h | 2.92 h |
| ImageReward | 1.50 h | 0.08 h | 0.67 h | 2.25 h | 2.55 h |
| Unified Reward | 1.51 h | 1.80 h | 0.67 h | 3.97 h | 4.03 h |

## C   Detailed Record of Computational Time

To facilitate a fair comparison between DPO and GRPO, as outlined in Sections 2.2, we maintain comparable training computational costs, measured in terms of computational time. The computational expense of DPO consists of three main components: (i) generating training images based on provided prompts, (ii) scoring these images using a reward model, and (iii) executing the subsequent

training phase. Detailed computational times for both GRPO and DPO are systematically recorded and presented in Table 6. Additionally, we assess and document the total training computational time for three key scaling strategies implemented for GRPO and DPO across different scaling ratios, as presented in Table 7. These computational time costs for both tables are evaluated using 8 A100 GPUs, with Janus-Pro [9] serving as the baseline.

Table 7: **Total Computational Time for Scaling Strategies Across Varying Scaling Ratios.**

| Scaling Strategy | Ratio 1 | | Ratio 2 | | Ratio 3 | |
|---|---|---|---|---|---|---|
| | DPO | GRPO | DPO | GRPO | DPO | GRPO |
| Data Scaling | 2.99 h | 2.92 h | 5.97 h | 5.84 h | 9.01 h | 8.76 h |
| Sampling Scaling | 2.99 h | 2.92 h | 5.33 h | 5.78 h | 7.66 h | 8.64 h |
| Iterative Scaling | 2.99 h | 2.92 h | 5.99 h | 5.84 h | 8.98 h | 8.76 h |

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
