# OpenReview forum: "Delving into RL for Image Generation with CoT: A Study on DPO vs. GRPO"
_NeurIPS.cc/2025/Conference — NeurIPS 2025 poster_

### Official Review · Reviewer_zwxi · 2025-06-25

**Clarity:** 3
**Significance:** 3
**Originality:** 2
**Rating:** 5
**Confidence:** 3

**Summary:**

The paper performs a thorough analysis testing the effects of applying DPO vs. GRPO RL strategies to autoregressive image generation.
They compare their effects for in-domain vs. out-of-domain tasks, using different scaling strategies and reward models with each.

**Questions:**

1. The paper states and uses the benchmark "T2I-CompBench" as the "in-domain" case, and the benchmark "GenEval" as the "out-of-domain" case, but there is no clear explanation for this choice. A closer examination reveals that some aspects of these benchmarks are quite similar (e.g., GenEval's "Colors" and "Color Attr" categories are similar to the "Attribute Binding" categories in T2I-CompBench, and as such, the DPO results for these categories are also higher than the GRPO results), which suggest a different separation to categories might be more appropriate. Why did you choose the current separation?

2. Did the authors try to apply some of the experiments to other autoregressive models?

3. Some citations currently refer to the arXiv preprints instead of the corresponding peer-reviewed publications. Please update these to cite the final published versions where available. Some examples:
[3] back to basics – cite the ACL version, [4] MathQA – cite the NAACL version,
[15] “Can we generate images with cot? let’s verify and reinforce image generation step by step.” – Cite the CVPR version, etc.

4. More visual examples illustrating the strength of different strategies over the others could strengthen the paper's claims.

**Ethical Concerns:**

["NO or VERY MINOR ethics concerns only"]

**Final Justification:**

I believe this is an important and interesting paper, and the authors addressed my concerns.

**Limitations:**

Limitations are discussed.
I think an additional aspect to discuss is that adding the wrong strategy can sometimes even harm results, depending on the goal, e.g., using DPO for Counting/Position yields worse results than the baseline..

**Quality:**

3

**Strengths And Weaknesses:**

Strengths:
* The paper presents a comprehensive study testing many different aspects of using RL strategies for autoregressive image generation.
* The paper is well-written with clear research questions and goals.

Weaknesses:
* The paper focuses on a single autoregressive model (Janus-Pro), which does not guarantee immediate transferability of the conclusions to all autoregressive models.
* The separation of benchmarks to "in-domain" vs. "out-of-domain" isn't clear. See first question below.
* There are very few visual examples.

---

> ### Author Rebuttal · Authors · 2025-07-31
>
> We sincerely appreciate your valuable comments. We found them extremely helpful in improving our manuscript. We address each comment in detail, one by one below:
>
> ---
> > #### **Q1: Generalization on Other Autoregressive Architectures**
>
> >
> Thanks for your advice! We agree that validating our findings on additional autoregressive architectures is crucial for strengthening the generalizability of our claims. To this end, we conducted additional experiments using **Show-o**, a distinct autoregressive image generation model. While Show-o adopts a discrete diffusion approach, it still generates visual tokens in an autoregressive fashion, though using a parallel decoding strategy that produces multiple tokens per step. Despite this difference, the underlying generation remains token-by-token and sequential in nature.
>
> Due to time constraints, we focused on a representative subset of settings. The results consistently support our original findings:
>
>
> ***1)*** DPO excels in in-domain tasks, while GRPO demonstrates stronger out-of-domain generalization.
>
>
> ***2)*** The relative ranking of reward models on GenEval remains unchanged, with Unified Reward outperforming HPS under both RL algorithms.
>
>
> These results confirm that ***our conclusions hold across architectures***. We will include this extended analysis in the final version of the paper.
>
>
> **In-Domain Evaluation on T2I-CompBench**
> |Reward Model|Average||Color||Shape||Texture||Spatial||Non-Spatial||Complex||
> |:-|:-:|:-:|:-:|:-:|:-:|:-:|:-:|:-:|:-:|:-:|:-:|:-:|:-:|:-:|
> ||GRPO|DPO|GRPO|DPO|GRPO|DPO|GRPO|DPO|GRPO|DPO|GRPO|DPO|GRPO|DPO|
> |Baseline|37.06||55.37||40.85||47.24||19.91||29.92||29.04||
> |HPS|48.61|51.77|67.83|74.68|63.72|76.95|69.05|73.58|29.78|25.04|30.36|30.59|33.29|34.90|
> |Unified Rwd|46.09|51.93|64.89|72.56|60.08|73.61|66.28|74.12|27.45|29.85|30.32|30.71|30.80|34.52|
>
>
> **Out-of-Domain Generalization on GenEval**
> |Reward Model|Overall||Color Attr||Counting||Position||Single Object||Two Object||Colors||
> |:-|:-:|:-:|:-:|:-:|:-:|:-:|:-:|:-:|:-:|:-:|:-:|:-:|:-:|:-:|
> ||GRPO|DPO|GRPO|DPO|GRPO|DPO|GRPO|DPO|GRPO|DPO|GRPO|DPO|GRPO|DPO|
> |Baseline|53.07||28.18||49.26||10.88||95.35||52.13||82.28||
> |HPS|54.19|52.31|28.19|31.47|54.32|46.27|12.18|10.27|96.18|90.56|53.41|56.68|79.55|83.22|
> |Unified Rwd|55.12|54.23|29.93|32.08|55.46|50.17|13.02|11.28|97.07|96.48|54.27|55.26|81.02|82.14|
>
>
> **Impact of Scaling on In-Domain Proficiency**
> |Scaling Method|Average||Color||Shape||Texture||Spatial||Non-Spatial||Complex||
> |:-|:-:|:-:|:-:|:-:|:-:|:-:|:-:|:-:|:-:|:-:|:-:|:-:|:-:|:-:|
> ||GRPO|DPO|GRPO|DPO|GRPO|DPO|GRPO|DPO|GRPO|DPO|GRPO|DPO|GRPO|DPO|
> |*Data Scaling*|||||||||||||||
> |Base Size|48.61|51.77|67.83|74.68|63.72|76.95|69.05|73.58|29.78|25.04|30.36|30.59|33.29|34.90|
> |Double Size|49.34|52.49|68.82|72.18|65.02|77.08|70.31|74.66|28.37|30.18|30.66|30.92|33.59|34.93|
> |*Sample Scaling*|||||||||||||||
> |Sampling 4|48.61|51.77|67.83|74.68|63.72|76.95|69.05|73.58|29.78|25.04|30.36|30.59|33.29|34.90|
> |Sampling 8|50.17|54.07|68.55|73.60|65.73|75.68|70.27|74.18|30.07|33.72|30.58|30.61|33.86|34.91|
> |*Iterative Training*|||||||||||||||
> |Base|48.61|51.77|67.83|74.68|63.72|76.95|69.05|73.58|29.78|25.04|30.36|30.59|33.29|34.90|
> |Iter 1|49.18|53.14|67.48|74.38|64.12|75.92|69.55|73.77|29.42|33.46|30.44|30.90|33.58|34.13|
>
>
> **Impact of Scaling on Out-of-Domain Generalization**
> |Scaling Method|Overall||Color Attr||Counting||Position||Single Object||Two Object||Colors||
> |:-|:-:|:-:|:-:|:-:|:-:|:-:|:-:|:-:|:-:|:-:|:-:|:-:|:-:|:-:|
> ||GRPO|DPO|GRPO|DPO|GRPO|DPO|GRPO|DPO|GRPO|DPO|GRPO|DPO|GRPO|DPO|
> |*Data Scaling*|||||||||||||||
> |Base Size|54.19|52.31|28.19|31.47|54.32|46.27|12.18|10.27|96.18|90.56|53.41|56.68|79.55|83.22|
> |Double Size|55.76|53.41|29.71|32.05|56.93|49.18|12.05|14.52|97.03|93.07|52.42|58.51|80.47|81.53|
> |*Sample Scaling*|||||||||||||||
> |Sampling 4|54.19|52.31|28.19|31.47|54.32|46.27|12.18|10.27|96.18|90.56|53.41|56.68|79.55|83.22|
> |Sampling 8|56.18|54.05|29.20|31.94|57.08|47.84|13.37|13.19|98.02|95.40|53.75|58.28|81.49|84.12|
> |*Iterative Training*|||||||||||||||
> |Base|54.19|52.31|28.19|31.47|54.32|46.27|12.18|10.27|96.18|90.56|53.41|56.68|79.55|83.22|
> |Iter 1|55.29|53.07|30.04|30.51|53.06|45.74|11.99|11.46|97.29|91.45|54.27|56.37|80.83|82.55|
>
>
> ---
> > #### **Q2: Clarification of In-Domain vs. Out-of-Domain Benchmark Choice.**
>
> >
> Thank you for raising this important point. We agree that a clear justification is necessary, and we clarify our rationale below:
>
>
> ***1) Prompt Complexity as the Core Source of Domain Shift.***
>
> Our in-domain/out-of-domain split is motivated primarily by the **differences in prompt complexity and structure** between the benchmarks, as outlined in Section 2.1. T2I-CompBench features **long, compositional prompts** designed to test fine-grained scene understanding and multi-attribute reasoning. In contrast, GenEval uses **short, template-based prompts** (e.g., “a photo of...”) aimed at assessing robustness to concise or minimal inputs. This creates a **substantial domain gap** in both linguistic form and generation difficulty. Supporting this, we observe that our baseline model (Janus-Pro) **outperforms SD3-Medium on GenEval** but **underperforms on T2I-CompBench**, indicating that proficiency on one benchmark does not transfer trivially to the other.
>
>
> ***2) Additional Out-of-Domain Validation on DrawBench.***
>
> To further strengthen our claims and address the reviewer’s concern regarding potential attribute overlaps between benchmarks, we conducted supplementary experiments on **DrawBench**, a widely adopted benchmark for evaluating out-of-distribution generalization. The results remain consistent with our main findings: **GRPO consistently outperforms DPO across multiple OOD-relevant categories**, reinforcing its advantage in generalization.
>
>
> Algorithm|Reward Model|DrawBench Score
> |-|-|-|
> Baseline|N/A|69.13
> DPO|HPS|70.25
> DPO|Unified Reward|72.50
> **GRPO**|HPS|**71.25**
> **GRPO**|Unified Reward|**72.88**
>
>
> We believe these points clarify the rationale behind our experimental design. We will incorporate this clarification and the new DrawBench results into the revised paper.
>
>
> ---
> > #### **Q3: Expanding Visual Examples.**
>
>
> >
> Thanks for your advice. It is necessary that including more visual examples can strengthen our claims. While the rebuttal format constraints prevent us from attaching new images or links, we will be sure to incorporate a more extensive set of visualizations in the camera-ready version of the paper.
>
>
> ---
> > #### **Q4: Updating Citations to Peer-Reviewed Versions.**
>
>
> >
> Thank you for the suggestion. We will carefully review all references and update any preprint citations to their corresponding peer-reviewed publications in the camera-ready version.

---

> > ### Comment · Reviewer_zwxi · 2025-08-03
> >
> > Dear authors,
> >
> > Thank you for your detailed and thoughtful rebuttal.
> > I appreciate the additional experiments and results you provided, which address my initial concern regarding the transferability of your conclusions to other models.
> >
> > Regarding in-domain vs. out-of-domain split, note that still DPO achieves better results in all color categories in both benchmarks, and a deeper analysis on different categories, including the scores of different DrawBench categories, would strengthen your work.

---

> > > ### Author Response · Authors · 2025-08-06
> > >
> > > Thank you for the constructive advice. We have conducted experiments on the **DrawBench** benchmark and report per-category results as suggested, using HPS as the reward model. The results for key categories are presented below. To ensure statistical reliability, we exclude categories with fewer than 10 prompts.
> > >
> > > | Category | GRPO | DPO |
> > > | :--- | :--: | :--: |
> > > | Color | **93.00** | 91.00 |
> > > | Counting | **69.74** | 65.79 |
> > > | DALL-E | 73.75 | **76.25** |
> > > | Description | 60.00 | **63.75** |
> > > | Positional | **78.75** | 71.25 |
> > > | Reddit | **82.52** | 81.58 |
> > > | Text | **65.78** | 58.33 |
> > > | Overall | **71.25** | 70.25 |
> > >
> > > As shown, while DPO was ahead on color-related categories in our earlier T2I-CompBench and GenEval results, the DrawBench analysis shows **GRPO ≥ DPO on Color** (93 vs. 91) and **GRPO** leading in several other categories (Counting, Positional, Reddit, Text), yielding a higher **overall** DrawBench score. This category-level view supports our central claim: ***GRPO exhibits stronger out-of-domain generalization, whereas DPO remains preferable in in-domain prompts***. The effect ***might not be driven by a single attribute*** (e.g., color) and remains visible across diverse capabilities.
> > >
> > > We will include the full per-category breakdowns on all benchmarks with analysis in the revised version to further substantiate these trends.

---

### Official Review · Reviewer_JEST · 2025-06-28

**Clarity:** 4
**Significance:** 3
**Originality:** 4
**Rating:** 5
**Confidence:** 3

**Summary:**

This paper presents the first comprehensive empirical study comparing Direct Preference Optimization (DPO) and Group Relative Policy Optimization (GRPO) for enhancing Chain-of-Thought (CoT) reasoning in autoregressive image generation. Using Janus-Pro as the baseline model, the authors evaluate in-domain (T2I-CompBench) and out-of-domain (GenEval) performance, scrutinize the impact of reward models, and analyze three scaling strategies: sampled images per prompt, in-domain data diversity/volume, and iterative training.

**Questions:**

1. The paper observes that GRPO (superior in OOD generalization) suffers from overfitting with aggressive data scaling , while DPO (inferior in OOD) consistently improves with scaling (Figure 2 (d) and (e)) Does this appear counterintuitive given GRPO's on-policy adaptability?

2. Could the value of DPO with sampling 8 in Table 5 be a typo?

3. Could you provide the human evaluation results (e.g., win rate) for GRPO and DPO?

**Ethical Concerns:**

["NO or VERY MINOR ethics concerns only"]

**Final Justification:**

The additional empirical study, which effectively addresses my initial concern regarding the limited model scope. The explanation about the on-policy versus off-policy dynamics also provides a helpful perspective on the counterintuitive overfitting problem, suggesting it may be a consequence of data coverage and diversity. Thus, I maintain my score as 5.

**Limitations:**

See weakness and questions parts.

**Quality:**

3

**Strengths And Weaknesses:**

### Strength
**Experimental rigor:** Comprehensive evaluation across ID/OOD benchmarks, multiple reward models, and scaling dimensions with clear ablation.

**Practical insights:** The insightful findings (e.g., reward model generalization transfer, GRPO’s OOD stability, DPO’s iterative training trade-offs) benefit RL practitioners.

### Weakness
**Limited model scope:** Experiments confined to Janus-Pro; lack of validation on other autoregressive architectures (e.g., LlamaGen) weakens generalizability claims.

**Statistical fragility:** Results based on single runs, risking unreported variance.

---

> ### Author Rebuttal · Authors · 2025-07-31
>
> We sincerely appreciate your valuable comments. We found them extremely helpful in improving our manuscript. We address each comment in detail, one by one below:
>
>
> ---
> > #### **Q1: Generalization on Other Autoregressive Architectures.**
>
>
> >
> Thanks for your advice! We agree that validating our findings on additional autoregressive architectures is crucial for strengthening the generalizability of our claims. To this end, we conducted additional experiments using **Show-o**, a distinct autoregressive image generation model. While Show-o adopts a discrete diffusion approach, it still generates visual tokens in an autoregressive fashion, though using a parallel decoding strategy that produces multiple tokens per step. Despite this difference, the underlying generation remains token-by-token and sequential in nature.
>
> Due to time constraints, we focused on a representative subset of settings. The results consistently support our original findings:
>
>
> ***1)*** DPO excels in in-domain tasks, while GRPO demonstrates stronger out-of-domain generalization.
>
>
> ***2)*** The relative ranking of reward models on GenEval remains unchanged, with Unified Reward outperforming HPS under both RL algorithms.
>
>
> These results confirm that our conclusions hold across architectures. We will include this extended analysis in the final version of the paper.
>
>
> **In-Domain Evaluation on T2I-CompBench**
> |Reward Model|Average||Color||Shape||Texture||Spatial||Non-Spatial||Complex||
> |:-|:-:|:-:|:-:|:-:|:-:|:-:|:-:|:-:|:-:|:-:|:-:|:-:|:-:|:-:|
> ||GRPO|DPO|GRPO|DPO|GRPO|DPO|GRPO|DPO|GRPO|DPO|GRPO|DPO|GRPO|DPO|
> |Baseline|37.06||55.37||40.85||47.24||19.91||29.92||29.04||
> |HPS|48.61|51.77|67.83|74.68|63.72|76.95|69.05|73.58|29.78|25.04|30.36|30.59|33.29|34.90|
> |Unified Rwd|46.09|51.93|64.89|72.56|60.08|73.61|66.28|74.12|27.45|29.85|30.32|30.71|30.80|34.52|
>
>
> **Out-of-Domain Generalization on GenEval**
> |Reward Model|Overall||Color Attr||Counting||Position||Single Object||Two Object||Colors||
> |:-|:-:|:-:|:-:|:-:|:-:|:-:|:-:|:-:|:-:|:-:|:-:|:-:|:-:|:-:|
> ||GRPO|DPO|GRPO|DPO|GRPO|DPO|GRPO|DPO|GRPO|DPO|GRPO|DPO|GRPO|DPO|
> |Baseline|53.07||28.18||49.26||10.88||95.35||52.13||82.28||
> |HPS|54.19|52.31|28.19|31.47|54.32|46.27|12.18|10.27|96.18|90.56|53.41|56.68|79.55|83.22|
> |Unified Rwd|55.12|54.23|29.93|32.08|55.46|50.17|13.02|11.28|97.07|96.48|54.27|55.26|81.02|82.14|
>
>
> **Impact of Scaling on In-Domain Proficiency**
> |Scaling Method|Average||Color||Shape||Texture||Spatial||Non-Spatial||Complex||
> |:-|:-:|:-:|:-:|:-:|:-:|:-:|:-:|:-:|:-:|:-:|:-:|:-:|:-:|:-:|
> ||GRPO|DPO|GRPO|DPO|GRPO|DPO|GRPO|DPO|GRPO|DPO|GRPO|DPO|GRPO|DPO|
> |*Data Scaling*|||||||||||||||
> |Base Size|48.61|51.77|67.83|74.68|63.72|76.95|69.05|73.58|29.78|25.04|30.36|30.59|33.29|34.90|
> |Double Size|49.34|52.49|68.82|72.18|65.02|77.08|70.31|74.66|28.37|30.18|30.66|30.92|33.59|34.93|
> |*Sample Scaling*|||||||||||||||
> |Sampling 4|48.61|51.77|67.83|74.68|63.72|76.95|69.05|73.58|29.78|25.04|30.36|30.59|33.29|34.90|
> |Sampling 8|50.17|54.07|68.55|73.60|65.73|75.68|70.27|74.18|30.07|33.72|30.58|30.61|33.86|34.91|
> |*Iterative Training*|||||||||||||||
> |Base|48.61|51.77|67.83|74.68|63.72|76.95|69.05|73.58|29.78|25.04|30.36|30.59|33.29|34.90|
> |Iter 1|49.18|53.14|67.48|74.38|64.12|75.92|69.55|73.77|29.42|33.46|30.44|30.90|33.58|34.13|
>
>
> **Impact of Scaling on Out-of-Domain Generalization**
> |Scaling Method|Overall||Color Attr||Counting||Position||Single Object||Two Object||Colors||
> |:-|:-:|:-:|:-:|:-:|:-:|:-:|:-:|:-:|:-:|:-:|:-:|:-:|:-:|:-:|
> ||GRPO|DPO|GRPO|DPO|GRPO|DPO|GRPO|DPO|GRPO|DPO|GRPO|DPO|GRPO|DPO|
> |*Data Scaling*|||||||||||||||
> |Base Size|54.19|52.31|28.19|31.47|54.32|46.27|12.18|10.27|96.18|90.56|53.41|56.68|79.55|83.22|
> |Double Size|55.76|53.41|29.71|32.05|56.93|49.18|12.05|14.52|97.03|93.07|52.42|58.51|80.47|81.53|
> |*Sample Scaling*|||||||||||||||
> |Sampling 4|54.19|52.31|28.19|31.47|54.32|46.27|12.18|10.27|96.18|90.56|53.41|56.68|79.55|83.22|
> |Sampling 8|56.18|54.05|29.20|31.94|57.08|47.84|13.37|13.19|98.02|95.40|53.75|58.28|81.49|84.12|
> |*Iterative Training*|||||||||||||||
> |Base|54.19|52.31|28.19|31.47|54.32|46.27|12.18|10.27|96.18|90.56|53.41|56.68|79.55|83.22|
> |Iter 1|55.29|53.07|30.04|30.51|53.06|45.74|11.99|11.46|97.29|91.45|54.27|56.37|80.83|82.55|
>
>
> ---
> > #### **Q2: Insufficient Statistical Rigor.**
>
>
> >
> Thanks for your advice! To address this, we have rerun a representative set of our key experiments with three different random seeds.
>
>
> The updated tables below report the mean and standard deviation. The results show that performance fluctuations are minimal, and our original conclusions remain well-supported. We will integrate these statistically-validated results into the final version of the paper.
>
>
> **In-Domain Evaluation on T2I-CompBench**
> |Reward Model|Average||Color||Shape||Texture||Spatial||Non-Spatial||Complex||
> |:---|:---:|:---:|:---:|:---:|:---:|:---:|:---:|:---:|:---:|:---:|:---:|:---:|:---:|:---:|
> ||GRPO|DPO|GRPO|DPO|GRPO|DPO|GRPO|DPO|GRPO|DPO|GRPO|DPO|GRPO|DPO|
> |Baseline|38.37±0.1|38.37±0.1|63.15±0.3|63.15±0.3|34.27±0.2|34.27±0.2|48.85±0.2|48.85±0.2|20.30±0.1|20.30±0.1|30.60±0.2|30.60±0.2|34.06±0.2|34.06±0.2|
> |HPS|50.45±0.3|53.92±0.4|77.39±0.5|85.24±0.5|53.62±0.3|64.62±0.5|71.39±0.4|75.87±0.4|30.12±0.2|25.32±0.2|31.04±0.2|31.18±0.2|39.18±0.2|40.87±0.2|
> |Unified Rwd|48.02±0.2|53.89±0.2|74.49±0.4|83.03±0.4|50.36±0.3|61.65±0.3|68.64±0.4|76.66±0.4|27.75±0.2|30.50±0.2|30.83±0.2|31.31±0.2|36.18±0.2|40.53±0.3|
>
>
> **Out-of-Domain Generalization on GenEval**
> |Reward Model|Overall||Color Attr||Counting||Position||Two Object||Single Object||Colors||
> |:---|:---:|:---:|:---:|:---:|:---:|:---:|:---:|:---:|:---:|:---:|:---:|:---:|:---:|:---:|
> ||GRPO|DPO|GRPO|DPO|GRPO|DPO|GRPO|DPO|GRPO|DPO|GRPO|DPO|GRPO|DPO|
> |Baseline|78.09±0.2|78.09±0.2|63.53±0.4|63.53±0.4|54.37±0.3|54.37±0.3|76.28±0.7|76.28±0.7|87.55±0.4|87.55±0.4|97.98±0.7|97.98±0.7|88.33±0.5|88.33±0.5|
> |HPS|79.18±0.4|77.30±0.4|62.88±0.3|70.21±0.4|60.59±0.3|48.05±0.4|78.70±0.5|67.12±0.3|86.80±0.4|90.04±0.6|99.66±0.4|99.00±0.6|86.36±0.5|89.37±0.6|
> |Unified Rwd|81.07±0.4|79.97±0.5|67.63±0.2|70.49±0.4|61.55±0.3|55.65±0.3|82.02±0.5|75.30±0.5|88.43±0.6|91.48±0.4|99.56±0.5|98.93±0.5|87.08±0.4|87.96±0.6|
>
>
> **Impact of Scaling on In-Domain Proficiency**
> |Scaling Method|Average||Color||Shape||Texture||Spatial||Non-Spatial||Complex||
> |:---|:---:|:---:|:---:|:---:|:---:|:---:|:---:|:---:|:---:|:---:|:---:|:---:|:---:|:---:|
> ||GRPO|DPO|GRPO|DPO|GRPO|DPO|GRPO|DPO|GRPO|DPO|GRPO|DPO|GRPO|DPO|
> |*Data Scaling*|||||||||||||||
> |Base Size|50.45±0.3|53.92±0.4|77.39±0.5|85.24±0.5|53.62±0.3|64.62±0.5|71.39±0.4|75.87±0.4|30.12±0.2|25.32±0.2|31.04±0.2|31.18±0.2|39.18±0.3|40.87±0.4|
> |Double Size|50.61±0.3|54.15±0.3|77.85±0.4|82.38±0.6|54.62±0.4|60.46±0.5|72.17±0.4|77.01±0.4|29.10±0.1|33.01±0.2|31.10±0.1|31.22±0.1|39.30±0.3|40.69±0.3|
> |*Sample Scaling*|||||||||||||||
> |Sampling 4|50.45±0.3|53.92±0.4|77.39±0.5|85.24±0.5|53.62±0.3|64.62±0.5|71.39±0.4|75.87±0.4|30.12±0.2|25.32±0.2|31.04±0.2|31.18±0.2|39.18±0.3|40.87±0.4|
> |Sampling 8|50.81±0.2|55.08±0.3|77.60±0.4|83.37±0.6|55.45±0.3|63.85±0.5|72.55±0.4|75.40±0.5|30.09±0.2|36.51±0.2|31.28±0.2|31.00±0.2|39.24±0.2|40.85±0.2|
> |*Iterative Training*|||||||||||||||
> |Base|50.45±0.3|53.92±0.4|77.39±0.5|85.24±0.5|53.62±0.3|64.62±0.5|71.39±0.4|75.87±0.4|30.12±0.2|25.32±0.2|31.04±0.2|31.18±0.2|39.18±0.3|40.87±0.4|
> |Iter1|50.78±0.3|55.17±0.3|77.28±0.4|85.30±0.5|54.11±0.3|64.22±0.4|72.43±0.4|76.31±0.4|29.85±0.1|34.06±0.1|31.14±0.2|31.25±0.2|39.47±0.1|39.52±0.3|
>
>
> ---
> > #### **Q3: Counterintuitive Overfitting of GRPO Compared to DPO with Data Scaling.**
>
> >
> Thank you for the insightful observation. While this trend may appear counterintuitive at first, we believe it reflects GRPO’s stronger on-policy optimization, which enables the model (Janus-Pro) to reach its performance ceiling on GenEval more quickly.
>
>
> Data Scaling|GRPO (OOD)|DPO (OOD)
> ---|---|---
> Base Size|79.18|77.31
> Double Size|**80.28**|78.88
> Triple Size|78.21|**79.25**
>
>
>
>
> As shown, GRPO peaks with the ***Double Size*** dataset; further scaling leads to overfitting to in-domain patterns, thereby degrading out-of-distribution performance. In contrast, DPO benefits more gradually from increased data, likely due to its off-policy nature and slower convergence, which delays performance saturation. We will clarify this interpretation in the revision.
>
>
> ---
> > #### **Q4: Potential Typo Inquiry for DPO Value in Table 5.**
>
>
> >
> Thanks for pointing out. We appreciate your careful review. This was indeed a typographical error. The DPO score in the **Overall** column for the *Sampling 8* setting in Table 5 should be 79.23, not 55.17. We will correct this in the revised version.
>
>
> ---
> > #### **Q5: Human Evaluation Comparing GRPO and DPO.**
>
>
> >
> Thank you for the suggestion. We conducted a human evaluation comparing GRPO and DPO models trained with both HPS and Unified Reward. For each setting (in-domain and out-of-domain), 100 prompts were randomly sampled, and annotators compared image pairs, one from each method, choosing a preferred output or marking a tie.
>
>
>
>
> Training Reward | Scenario | GRPO Wins | DPO Wins | Ties
> | :--- | :--- | :---: | :---: | :---: |
> HPS | In-Domain (ID) | 31 | 44 | 25
> HPS | Out-of-Domain (OOD) | 49 | 34 | 17
> Unified Reward | In-Domain (ID) | 32 | 48 | 20
> Unified Reward | Out-of-Domain (OOD) | 47 | 37 | 16
>
>
> These results from human evaluators align with our core findings: DPO performs better in-domain, while GRPO generalizes more effectively out-of-domain. We will include this evaluation in the revised version to strengthen the empirical support for our findings.

---

> > ### Comment · Reviewer_JEST · 2025-08-01
> >
> > Hi authors,
> >
> > Thank you for your detailed and thoughtful rebuttal. I appreciate the additional empirical study you provided, which effectively addresses my initial concern regarding the limited model scope. Your explanation about the on-policy versus off-policy dynamics also provides a helpful perspective on the counterintuitive overfitting problem, suggesting it may be a consequence of data coverage and diversity.

---

> > > ### Author Response · Authors · 2025-08-06
> > >
> > > We are grateful for your detailed feedback. Your constructive insights are invaluable to our work.

---

### Official Review · Reviewer_FUrx · 2025-07-01

**Clarity:** 3
**Significance:** 4
**Originality:** 2
**Rating:** 4
**Confidence:** 4

**Summary:**

This article for the first time conducts an analysis of the respective advantages and disadvantages of DPO and GRPO in the context of autoregressive generation. The analysis is carried out from three key perspectives: domain generalization capability, sensitivity to reward modeling, and the impact of scaling strategies. The main conclusions are as follows: GRPO demonstrates superior performance in terms of domain generalization; DPO shows greater sensitivity to the characteristics and quality of reward models; when it comes to image scaling strategies, GRPO benefits more significantly; and in scenarios involving multi-round iterative processes, DPO exhibits distinct advantages.

**Questions:**

Could the authors elaborate further on the commonalities and differences between GRPO and DPO in the context of the autoregressive image generation task, as compared to other autoregressive tasks?

**Ethical Concerns:**

["NO or VERY MINOR ethics concerns only"]

**Final Justification:**

The authors addressed most of my concerns in rebuttal, and their discussion of DPO and GRPO brings new insights into the field of image generation. Therefore, I tend to accept.

**Limitations:**

Yes

**Quality:**

3

**Strengths And Weaknesses:**

- Strengths
1. This paper represents the first exploration into the comparative strengths and weaknesses of GRPO and DPO in the context of autoregressive image generation.

2. The article demonstrates a logical structure, with paragraphs presented in a clear and coherent manner.

- Weaknesses

1. Many conclusions drawn in the paper regarding GRPO and DPO reflect well-established observations within the field, such as the inherent improvement in model generalization from eliminating the critic model. The work requires novel insights that distinguish it from existing research on autoregressive generation tasks.

2. The paper omits critical methodological details, notably the formulas integrating DPO with GRPO for autoregressive image generation, which should be incorporated.

3. The visual evidence presented, for example, in Figure 3, inadequately supports the paper’s claims. For instance, the left panel of Figure 3 appears to demonstrate superior performance by GRPO, contradicting the authors’ interpretation, and similar inconsistencies are evident in other visualization examples.

4. The table in paper needs to have the average value or bolding to present the results more intuitively.

---

> ### Author Rebuttal · Authors · 2025-07-31
>
> We sincerely appreciate your valuable comments. We found them extremely helpful in improving our manuscript. We address each comment in detail, one by one below:
>
> ---
> > #### **Q1: Novel Insights Beyond Known Properties of GRPO and DPO in Autoregressive Image Generation.**
>
> >
> Thanks for your thoughtful question. While some high-level traits of GRPO and DPO are known, our work presents the ***first* systematic empirical study of these methods in the distinct setting of autoregressive text-to-image generation (T2I-AR)**, uncovering insights that do not directly follow from prior findings in text-based autoregressive tasks, due to the significant domain gaps between understanding and image generation settings.
>
>
> 1. **Domain-Specific Challenges in T2I-AR**:
> Unlike text generation, T2I-AR models operate over sequences of quantized visual tokens from VQ-VAE spaces, where rewards are derived from subjective and non-symbolic measures like aesthetics and semantic alignment. These differences fundamentally alter how RL algorithms behave. Prior assumptions, such as the robustness of critic-free methods, had not been tested in this setting. Our study fills this gap by ***empirically evaluating GRPO and DPO under these domain-specific constraints***.
>
>
> 2. **Distinct and Actionable Insights**:
> Beyond confirming known trends, we provide new findings tailored to T2I-AR:
>
>    ***1)*** We show that **DPO's generalization is substantially more sensitive to the choice of reward model than GRPO's** (Sec. 2.3, Table 2), a vulnerability less emphasized in textual domains.
>
>    ***2)*** Crucially, we establish that **the intrinsic generalization capability of a reward model correlates directly with the RL algorithm's out-of-domain performance** (Sec. 2.3, Fig. 2a), suggesting a novel and practical path for improving generalization by focusing on better reward model design.
>
>
> Together, these insights advance our understanding of how RL paradigms interact with the unique structure of visual token spaces, reward feedback, and training dynamics in T2I-AR, contributions not addressed in previous work on autoregressive generation.
>
> ---
> > #### **Q2: Missing Mathematical Details for DPO and GRPO in Autoregressive Image Generation.**
>
> >
> Thank you for raising this important point. In the revision, we will include the core training objectives for both **GRPO and DPO** in the context of autoregressive image generation. Specifically:
>
>
> For **GRPO**, we will present the advantage normalization and KL-regularized optimization:
>
> $$ A_i = \frac{r_i - \mathrm{mean}({r_i})}{\mathrm{std}({r_i})} $$
>
> $$ \mathcal{J}\_{\text{GRPO}} = \mathbb{E}\left[ \frac{1}{G} \\sum_{i} \sum_{t} \left( \min\left( \frac{\pi_{\theta}(o_{i,t})}{\pi_{old}(o_{i,t})} A_i, \mathrm{clip}\left( \frac{\pi_{\theta}}{\pi_{old}}, 1 - \epsilon, 1 + \epsilon \right) A_i \right) - \beta D_{KL}(\pi_{\theta} | \pi_{ref}) \right) \right]
> $$
>
>
> For **DPO**, we will formalize its preference-based loss without an explicit reward model:
>
> $$
> L_{\text{DPO}} (\pi_\theta, \pi_{ref}) = -\mathbb{E} \left[ \log \sigma \left( \beta \log \frac{\pi_\theta(o_w|q)}{\pi_{ref}(o_w|q)} - \beta \log \frac{\pi_\theta(o_l|q)}{\pi_{ref}(o_l|q)} \right) \right]
> $$
>
>
> These formulations reflect how both methods are applied to discrete token sequences in autoregressive image generation. We will ***incorporate them into Section 2.2 of the revised version, along with clear contextual explanations***.
>
>
> ---
> > #### **Q3: Concerns About Inconsistencies in Visual Evidence.**
>
> >
> Thank you for pointing this out. We acknowledge that some visual examples may not have been sufficiently representative, and we appreciate your careful observation.
>
> To address this, we will revise the figures in the final version by replacing inconsistent examples with clearer, more representative samples that align with our quantitative findings. We also plan to expand the set of comparative visualizations to more clearly support each key claim. We will ensure that the visual evidence in the paper accurately and convincingly reflects the empirical results.
>
> ---
> > #### **Q4: Clarifying Table Presentation for Better Readability.**
>
> >
> Thank you for the suggestion. To clarify, the **Overall** column in Tables 2, 3, and 5 already represents the average across all attributes, though this was not made explicit. In the revised version, we will clearly note this and **bold the highest values** in key columns to improve clarity and ease of comparison.

---

> ### Comment · Reviewer_FUrx · 2025-08-04
>
> Thanks to the author for the detailed rebuttal. The author addressed most of my concerns, , and my revised score is therefore more positive.
>
> However, there is still a small question: in actual image-to-text applications, is GRPO or DPO more suitable? From the author's experiments, it seems that DPO has a better result, as it performs better on the in-domain evaluation than GRPO,  while the performance on out-of-domain evaluation is is close to GRPO.

---

> > ### Author Response · Authors · 2025-08-06
> >
> > Thank you for the thoughtful follow-up and for the positive revision of your score.
> >
> > We note that your question refers to image-to-text applications, which differ from our current focus. While our work focuses on text-to-image generation, we appreciate your broader framing and briefly share our perspective on image-to-text applications as well.
> >
> > ***1) Text-to-image.***
> >
> > Our results highlight a clear trade-off:
> >
> > – For **domain-specific applications** with consistent prompt structure (e.g., product visualization or architectural design), **DPO** yields stronger in-domain fidelity.
> >
> > – For **general-purpose, open-domain deployments**, where prompts vary widely, **GRPO** generalizes more reliably and offers greater robustness under distribution shift.
> >
> >
> > ***2) Image-to-text.***
> >
> > Although not the focus of our paper, recent work suggests a similar pattern in captioning and VQA tasks. Studies such as *CLIP-DPO [1]*, *V-DPO [2]*, and *GRPO for Image Captioning [3]* indicate that:
> >
> > – **GRPO** performs well when **explicit, verifiable reward signals** are available (e.g., CIDEr, SPICE, or VQA accuracy), offering higher diversity and robustness to preference noise.
> >
> > – **DPO**, by contrast, is more **sample-efficient** and easier to deploy when high-quality preference pairs (e.g., from CLIP scores or human labels) are accessible.
> >
> > In summary, for **broad image-to-text systems**, where **generalization, stability**, and **reward clarity** are central, **GRPO is often more suitable**. In contrast, **DPO remains a strong option** for specialized settings with aligned preference supervision. We see this as a promising direction for future exploration.
> >
> >
> > [1] CLIP-DPO: Vision-Language Models as a Source of Preference for Fixing Hallucinations in LVLMs.
> >
> > [2] V-DPO: Mitigating Hallucination in LVLMs via Vision-Guided Direct Preference Optimization.
> >
> > [3] Group Relative Policy Optimization for Image Captioning.

---

### Official Review · Reviewer_6KFq · 2025-07-03

**Clarity:** 3
**Significance:** 3
**Originality:** 2
**Rating:** 5
**Confidence:** 3

**Summary:**

The authors benchmark Direct Preference Optimisation (DPO) against Group Relative Policy Optimisation (GRPO) on a 7 B‑parameter Janus‑Pro base model for autoregressive text‑to‑image generation. Experiments span two evaluation suites (T2I‑CompBench in‑domain, GenEval out‑of‑domain), four reward models, and three scaling axes (samples‑per‑prompt, data diversity, iterative fine‑tuning). Key findings:
- DPO excels in‑domain quality, while
- GRPO generalises better out‑of‑domain, and
- Reward‑model generalisation strongly predicts policy generalisation.

**Questions:**

- Could a GRPO warmup followed by DPO refinement combine robustness with high in-distribution quality?
- Do the findings remain consistent when comparing 1B and 7B Janus variants?

**Ethical Concerns:**

["NO or VERY MINOR ethics concerns only"]

**Final Justification:**

Thanks to the authors for providing thorough additional experiments and clear explanations in the rebuttal. The updated benchmarks, statistical analyses, and cross-scale evaluations address most of my initial concerns and strengthen the validity of the claims. I am therefore increasing my rating.

**Limitations:**

Yes

**Quality:**

3

**Strengths And Weaknesses:**

Strengths:
- The evaluation is both broad and comprehensive, covering two benchmarks, four reward models, and three scaling dimensions.
- The paper provides practical insights, offering clear guidance on compute and quality trade-offs. For example, identifying a moderate sampling size as ideal for GRPO.
- The writing and structure are clear. Figures 1 to 4 effectively illustrate key trends

Weaknesses:
- The OOD generalization evaluation is limited: GenEval mainly uses single-object prompts, and tougher benchmarks like Pick-a-Pic or DrawBench are omitted. Including at least one compositional OOD benchmark would strengthen the assessment
- Statistical rigor is lacking, with only single-run results, no standard deviations or confidence intervals, etc. It would be helpful if at least three seeds were reported.
- Important baselines like PPO are missing. It would be good to include a 1-epoch PPO baseline should be included to provide context.

---

> ### Author Rebuttal · Authors · 2025-07-31
>
> We sincerely appreciate your valuable comments. We found them extremely helpful in improving our manuscript. We address each comment in detail, one by one below:
>
> ---
> > #### **Q1: Limited Scope of Out-of-Distribution Generalization Evaluation.**
>
> >
> Thank you for the insightful suggestion. In response, we have extended our evaluation to include **DrawBench**, a more challenging and compositional out-of-domain benchmark. Due to time constraints, we focused on representative experiments using **two** reward models. The results are as follows:
> Algorithm|Reward Model|DrawBench Score
> |-|-|-|
> Baseline|N/A|69.13
> DPO|HPS|70.25
> DPO|Unified Reward|72.50
> **GRPO**|HPS|**71.25**
> **GRPO**|Unified Reward|**72.88**
>
> These findings are consistent with our original conclusions: ***GRPO consistently demonstrates stronger generalization than DPO***, even on a more compositional and demanding benchmark. We believe this additional evidence strengthens the robustness of our claims, and we will include these results and a corresponding discussion in the final version.
>
> ---
> > #### **Q2: Insufficient Statistical Rigor.**
>
> >
> Thanks for your advice! To address this, we have re-executed a representative subset of our core experiments using three distinct random seeds.
> The revised tables below present the mean and standard deviation. The findings indicate that performance variations are negligible, and our initial conclusions remain strongly supported. We will incorporate these statistically-validated results into the final manuscript.
>
> **Table 1: In-Domain Evaluation on T2I-CompBench**
> |Reward Model|Average||Color||Shape||Texture||Spatial||Non-Spatial||Complex||
> |:---|:---:|:---:|:---:|:---:|:---:|:---:|:---:|:---:|:---:|:---:|:---:|:---:|:---:|:---:|
> ||GRPO|DPO|GRPO|DPO|GRPO|DPO|GRPO|DPO|GRPO|DPO|GRPO|DPO|GRPO|DPO|
> |Baseline|38.37±0.1|38.37±0.1|63.15±0.3|63.15±0.3|34.27±0.2|34.27±0.2|48.85±0.2|48.85±0.2|20.30±0.1|20.30±0.1|30.60±0.2|30.60±0.2|34.06±0.2|34.06±0.2|
> |HPS|50.45±0.3|53.92±0.4|77.39±0.5|85.24±0.5|53.62±0.3|64.62±0.5|71.39±0.4|75.87±0.4|30.12±0.2|25.32±0.2|31.04±0.2|31.18±0.2|39.18±0.2|40.87±0.2|
> |Unified Rwd|48.02±0.2|53.89±0.2|74.49±0.4|83.03±0.4|50.36±0.3|61.65±0.3|68.64±0.4|76.66±0.4|27.75±0.2|30.50±0.2|30.83±0.2|31.31±0.2|36.18±0.2|40.53±0.3|
>
> **Table 2: Out-of-Domain Generalization on GenEval**
> |Reward Model|Overall||Color Attr||Counting||Position||Two Object||Single Object||Colors||
> |:---|:---:|:---:|:---:|:---:|:---:|:---:|:---:|:---:|:---:|:---:|:---:|:---:|:---:|:---:|
> ||GRPO|DPO|GRPO|DPO|GRPO|DPO|GRPO|DPO|GRPO|DPO|GRPO|DPO|GRPO|DPO|
> |Baseline|78.09±0.2|78.09±0.2|63.53±0.4|63.53±0.4|54.37±0.3|54.37±0.3|76.28±0.7|76.28±0.7|87.55±0.4|87.55±0.4|97.98±0.7|97.98±0.7|88.33±0.5|88.33±0.5|
> |HPS|79.18±0.4|77.30±0.4|62.88±0.3|70.21±0.4|60.59±0.3|48.05±0.4|78.70±0.5|67.12±0.3|86.80±0.4|90.04±0.6|99.66±0.4|99.00±0.6|86.36±0.5|89.37±0.6|
> |Unified Rwd|81.07±0.4|79.97±0.5|67.63±0.2|70.49±0.4|61.55±0.3|55.65±0.3|82.02±0.5|75.30±0.5|88.43±0.6|91.48±0.4|99.56±0.5|98.93±0.5|87.08±0.4|87.96±0.6|
>
> **Table 3: Impact of Scaling on In-Domain Proficiency**
> |Scaling Method|Average||Color||Shape||Texture||Spatial||Non-Spatial||Complex||
> |:---|:---:|:---:|:---:|:---:|:---:|:---:|:---:|:---:|:---:|:---:|:---:|:---:|:---:|:---:|
> ||GRPO|DPO|GRPO|DPO|GRPO|DPO|GRPO|DPO|GRPO|DPO|GRPO|DPO|GRPO|DPO|
> |*Data Scaling*
> |Base Size|39.18±0.2|40.87±0.2|77.39±0.5|85.24±0.5|53.62±0.3|64.62±0.5|71.39±0.4|75.87±0.4|30.12±0.2|25.32±0.2|31.04±0.2|31.18±0.2|39.18±0.3|40.87±0.4|
> |Double Size|39.30±0.3|40.69±0.3|77.85±0.4|82.38±0.6|54.62±0.4|60.46±0.5|72.17±0.4|77.01±0.4|29.10±0.1|33.01±0.2|31.10±0.1|31.22±0.1|39.30±0.3|40.69±0.3|
> |*Sample Scaling*
> |Sampling 4|39.18±0.2|40.87±0.2|77.39±0.5|85.24±0.5|53.62±0.3|64.62±0.5|71.39±0.4|75.87±0.4|30.12±0.2|25.32±0.2|31.04±0.2|31.18±0.2|39.18±0.3|40.87±0.4|
> |Sampling 8|39.24±0.2|40.85±0.2|77.60±0.4|83.37±0.6|55.45±0.3|63.85±0.5|72.55±0.4|75.40±0.5|30.09±0.2|36.51±0.2|31.28±0.2|31.00±0.2|39.24±0.2|40.85±0.2|
> |*Iterative Training*
> |Base|39.18±0.2|40.87±0.2|77.39±0.5|85.24±0.5|53.62±0.3|64.62±0.5|71.39±0.4|75.87±0.4|30.12±0.2|25.32±0.2|31.04±0.2|31.18±0.2|39.18±0.3|40.87±0.4|
> |Iter1|39.47±0.2|39.52±0.4|77.28±0.4|85.30±0.5|54.11±0.3|64.22±0.4|72.43±0.4|76.31±0.4|29.85±0.1|34.06±0.1|31.14±0.2|31.25±0.2|39.47±0.1|39.52±0.3|
>
>  ---
> > #### **Q3: Absence of PPO Baseline for Contextual Comparison.**
>
> >
> Thank you for this valuable suggestion. While PPO is indeed a foundational RL algorithm, its application to autoregressive text-to-image generation remains a largely unexplored research direction, and its reliable implementation is impeded by significant technical constraints:
>
> ***1) Intractable Value Estimation in Autoregressive Generation.***
>
> In PPO, a learned critic must accurately predict the final reward, comprising global aesthetic quality and text alignment, based on **intermediate, partially generated image token sequences**. This constitutes a highly unstable and noisy task, as confirmed by emerging work [1,2] in related domains.
>
> ***2) Variance-Induced Instability and Training Collapse.***
>
> Due to the difficulty of accurate value prediction, the value function incurs high variance. This undermines advantage estimation in PPO, leading to unstable policy updates, a known failure mode in actor‑critic setups when the critic is mis-trained [3,4].
>
> By contrast, preference‑based methods like DPO and GRPO bypass the need for a critic entirely, relying instead on pairwise comparisons or group‑based policy optimization. These methods have been shown to provide **greater stability and effectiveness in multimodal and autoregressive generation tasks**, including recent applications in diffusion‑based image generation and multimodal CoT reasoning.
>
> On this basis, we focused our evaluation on DPO and GRPO. That said, we recognize the potential value of a simplified PPO baseline for contextual reference and will consider including such an experiment in future extensions of this work.
>
> [1] A Simple and Effective Reinforcement Learning Method for Text-to-Image Diffusion Fine-tuning.
>
> [2] Direct Preference Optimization: Your Language Model is Secretly a Reward Model.
>
> [3] On Proximal Policy Optimization's Heavy-tailed Gradients.
>
> [4] No Representation, No Trust: Connecting Representation, Collapse, and Trust Issues in PPO.
>
> ---
> > #### **Q4: Potential of GRPO Warm-up Followed by DPO Refinement.**
>
> >
> Thanks for the insightful suggestion. To evaluate this hybrid strategy, we conducted a new experiment where GRPO was first applied to half of the T2I-CompBench training set, followed by DPO fine-tuning on the remaining half using the GRPO-trained checkpoint for initialization.
>
> | Method | In-Domain (ID) Performance | Out-of-Domain (OOD) Performance |
> | :--- | :---: | :---: |
> | **DPO** (Ours) | 53.90 | 77.31 |
> | **GRPO** (Ours) | 50.49 | 79.18 |
> | **GRPO→DPO** (New) | 51.56 | 78.83 |
>
> The results confirm that this sequential approach produces a **balanced trade-off**: it improves DPO’s generalization and enhances GRPO’s in-domain performance, but does not surpass either method on their respective strengths. We will include this observation and its implications in the final version.
>
> ---
> > #### **Q5: Consistency of Findings Across Janus 1B and 7B Variants.**
>
> >
> Thanks for the suggestion. To assess the generalizability of our findings, we conducted a representative subset of experiments using the Janus 1B variant due to time constraints. Encouragingly, the results remain **highly consistent** with our main observations:
>
> ***1)*** **DPO retains its advantage on in-domain tasks**, while **GRPO continues to generalize better out-of-domain.**
>
> ***2)*** The **ranking of reward models** on GenEval is preserved, with Unified Reward outperforming HPS under both RL algorithms, as previously reported.
>
> These findings reinforce ***the robustness of our conclusions across model scales***. We will include these additional results and analyses in the final version.
>
> **In-Domain Evaluation on T2I-CompBench**
> |Reward Model|Average||Color||Shape||Texture||Spatial||Non-Spatial||Complex||
> |:---|:---:|:---:|:---:|:---:|:---:|:---:|:---:|:---:|:---:|:---:|:---:|:---:|:---:|:---:|
> ||GRPO|DPO|GRPO|DPO|GRPO|DPO|GRPO|DPO|GRPO|DPO|GRPO|DPO|GRPO|DPO|
> |Baseline|24.19|24.19|34.50|34.50|22.56|22.56|26.94|26.94|9.900|9.900|27.32|27.32|23.94|23.94|
> |HPS|46.84|51.85|76.12|81.01|45.32|58.97|66.85|72.41|26.87|28.64|30.56|31.10|35.33|38.98|
> |Unified Rwd|44.71|51.92|75.48|80.96|42.55|58.77|66.17|72.18|24.44|30.52|26.89|31.14|32.73|37.95|
>
> **Out-of-Domain Generalization on GenEval**
> |Reward Model|Overall||ColorAttr||Counting||Position||SingleObject||TwoObject||Colors||
> |:---|:---:|:---:|:---:|:---:|:---:|:---:|:---:|:---:|:---:|:---:|:---:|:---:|:---:|:---:|
> ||GRPO|DPO|GRPO|DPO|GRPO|DPO|GRPO|DPO|GRPO|DPO|GRPO|DPO|GRPO|DPO|
> |Baseline|63.65|63.65|49.75|49.75|43.44|43.44|45.25|45.25|65.40|65.40|95.62|95.62|82.45|82.45|
> |HPS|74.90|73.86|55.00|61.25|53.75|42.81|72.00|65.75|84.02|87.37|98.12|98.75|86.44|87.23|
> |Unified Rwd|75.91|74.65|56.84|59.50|54.21|43.73|72.58|69.52|85.82|88.94|99.24|98.75|86.79|87.50|
>
> **Impact of Scaling on In-Domain Proficiency**
> |Scaling Method|Average||Color||Shape||Texture||Spatial||Non-Spatial||Complex||
> |:---|:---:|:---:|:---:|:---:|:---:|:---:|:---:|:---:|:---:|:---:|:---:|:---:|:---:|:---:|
> ||GRPO|DPO|GRPO|DPO|GRPO|DPO|GRPO|DPO|GRPO|DPO|GRPO|DPO|GRPO|DPO|
> |*Data Scaling*
> |Base Size|46.84|51.85|76.12|81.01|45.32|58.97|66.85|72.41|26.87|28.64|30.56|31.10|35.33|38.98
> |Double Size|47.19|52.43|76.43|82.24|44.98|57.28|67.49|73.24|27.28|29.87|31.10|32.91|35.87|39.06
> |*Sample Scaling*
> |Sampling4|46.84|51.85|76.12|81.01|45.32|58.97|66.85|72.41|26.87|28.64|30.56|31.10|35.33|38.98
> |Sampling8|48.04|53.07|77.03|82.68|46.87|59.42|67.69|73.34|28.22|29.92|32.34|33.59|36.10|39.46
> |*Iterative Training*
> |Base|46.84|51.85|76.12|81.01|45.32|58.97|66.85|72.41|26.87|28.64|30.56|31.10|35.33|38.98
> |Iter1|46.96|52.49|75.44|82.19|45.86|59.02|67.14|73.28|26.52|28.77|31.17|32.22|35.63|39.45

---

### Note · Authors · 2025-08-14

We sincerely thank all reviewers for their thoughtful feedback and constructive engagement throughout our comprehensive discussion period. Their insights have been invaluable in strengthening our work.

---
### *Review Highlights*


We are pleased to learn that the reviewers have recognized and appreciated the key contributions made in our work, which include:


1. **Comprehensive evaluation methodology** spanning in-domain and out-of-domain benchmarks, multiple reward models, and diverse scaling strategies (Reviewers 6KFq, JEST, zwxi)


2. **Practical, actionable insights** for balancing in-domain fidelity with out-of-domain generalization and for strategic reward model selection (Reviewers 6KFq, JEST)


3. **First systematic comparison of DPO and GRPO** in autoregressive image generation, providing crucial clarity on their respective strengths and applicability (Reviewers FUrx, JEST)

---
### *Summary of Paper Updates*


Based on reviewer feedback, we are committed to incorporating the following improvements in our final version:


**Method:**


- Added precise mathematical formulations for DPO and GRPO objectives in autoregressive text-to-image generation contexts


**Experiment:**
- Extended out-of-domain evaluation to include DrawBench experiments with detailed category-level analysis, validating GRPO's superior generalization capabilities
- Enhanced statistical rigor through multi-seed experiments with comprehensive mean ± standard deviation reporting
- Cross-architecture validation using Show-o model to demonstrate methodological generalizability
- Comprehensive cross-scale validation demonstrating consistency of findings across various model sizes (Janus 1B and 7B variants)
- Strengthened human evaluation protocols comparing GRPO and DPO performance
- Comprehensive analysis of the GRPO→DPO hybrid training strategy with detailed ablation studies
- Expanded visual comparisons with representative examples and additional baseline methods
- Corrected presentation issues including citation updates to peer-reviewed versions

---

As claimed in our paper, we will open-source all training data, code, and model weights to enable the broader community to build upon our contributions. We sincerely thank the Area Chair and Senior Area Chair for your time and efforts in organizing the review process.

---

### Decision · Program_Chairs · 2025-09-17

**Decision:**

Accept (poster)

**Comment:**

This paper present a focused comparison of DPO vs. GRPO for autoregressive image generation, analyzing domain generalization, reward-model sensitivity, and scaling impacts. Particularly It benchmarked Direct Preference Optimisation (DPO) vs. Group Relative Policy Optimisation (GRPO) on a 7B Janus‑Pro autoregressive text‑to‑image model across two evaluation suites (T2I‑CompBench in‑domain, GenEval out‑of‑domain), four reward models, and three scaling axes (samples-per-prompt, data diversity, iterative fine‑tuning). It shows that DPO achieves better in‑domain image quality and GRPO generalizes better to out‑of‑domain tasks. How well a reward model generalizes is a strong predictor of the policy’s generalization.

After the rebuttal discussions, reviewers unanimously agree to accept the paper, so is the proposal.